# KNOWLEDGE GRAPH EMBEDDING: A PROBABILISTIC PERSPECTIVE AND GENERALIZATION BOUNDS

## ABSTRACT

We study theoretical properties of embedding methods for knowledge graph completion under the "missing completely at random" assumption. We prove generalization error bounds for this setting. Even though the missing completely at random setting may seem naive, it is actually how knowledge graph embedding methods are typically benchmarked in the literature. Our results provide, to a certain extent, an explanation for why knowledge graph embedding methods work (as much as classical learning theory results provide explanations for classical learning from i.i.d. data).

## 1 INTRODUCTION

Many large knowledge bases such as YAGO (Mahdisoltani et al., 2015) and NELL (Carlson et al., 2010) are represented as *knowledge graphs*. A knowledge graph is a set of triples $(head, predicate, tail)$ where *head* and $tail$ are *objects* (also frequently called *entities*) and *predicate* is a binary relation. Each such triple represents a fact about the world, e.g. $(NewYork, IsLocatedIn, US)$. The objects can be seen as vertices of a graph and the relations as labeled directed edges (pointing from head to tail), hence the name *knowledge* graph. Most knowledge bases are incomplete, i.e. there are facts that should be included in them but are not. A popular task is to automatically complete such knowledge bases. This is often done by embedding the objects and relations contained in them into some vector spaces and then using the embedding to predict the missing tuples.

There has been a number of knowledge graph completion methods presented in the literature (we give an overview of existing methods in Section 2.3). However, to our best knowledge, there has not been any theoretical analysis of the generalization abilities of these methods in the spirit of classical learning theory. We make the first steps in this direction in the present paper. We derive expected error bounds for a class of knowledge graph embedding methods. A key challenge in deriving generalization bounds in this setting is that the training and test data do not satisfy assumptions typically used in learning theory. The knowledge bases that we receive contain only positive examples, that is, the facts already in the knowledge base. We do not know which of the facts not contained in the knowledge base are just missing and which are not contained in the knowledge base because they are actually false. This is known as learning from positive and unlabeled data (PU-learning); we refer to (Bekker & Davis, 2018) for a recent survey. Only a few results have been obtained for generalization bounds for PU-learning so far, e.g. (Niu et al., 2016), but they are not directly applicable to our setting (we discuss the details in Section 10).

The bounds that we derive in this paper rely on an assumption on how the triples are missing from the knowledge graph. In particular, we assume that the triples are missing "completely at random", i.e. that each tuple may be missing with probability $\delta > 0$ independently of the others. Even though this setting may appear naive, it is actually exactly how some famous datasets' training and test sets are created, e.g. (Bordes et al., 2013). Thus, the theoretical results actually do relate to what one observes in "practice" (i.e. in the respective papers that use these datasets). The bounds that we derive here apply to knowledge graph embedding methods that use log-loss for learning the embeddings such as ComplEx (Trouillon & Nickel, 2017), SimplE (Kazemi & Poole, 2018) or ConvE (Dettmers et al., 2018).

**Main Technical Contributions** Our main technical results are expected error bounds on the number of triples $(h, r, t)$ that are predicted incorrectly when using knowledge graph embedding for knowledge graph completion (stated in Theorems 4 and 5), i.e. either to be in the knowledge graph when they are not supposed to be (false positives) or not to be in the knowledge graph when they are supposed to be (false negatives). The bound assumes that the knowledge graph embedding is learned by maximizing the log-likelihood of a certain exponential-family distribution (introduced in Section 5), which turns out to be equivalent to how several prominent knowledge graph embedding methods (Trouillon & Nickel, 2017; Kazemi & Poole, 2018; Dettmers et al., 2018) are learned using log-loss (this is discussed in detail in Section 10). Our results therefore provide a statistical learning theory style justification for these knowledge graph embedding methods. In order to obtain the main results we also derive several auxiliary technical results which might be of independent interest (in particular, Theorems 2 and 3).

## 2 PRELIMINARIES

### 2.1 KNOWLEDGE GRAPHS

Let $\mathcal{O}$ be a set of objects and $\mathcal{R}$ a set of relations. A knowledge graph is a set of triples $(h, r, t)$ where $h$ and $t$ are objects and $r$ is a relation. Here $h$ is called the *head* of the triple and $t$ is called the *tail* of the triple. In the formalism of first-order logic, a triple $(h, r, t)$ represents a first-order logic atom $r(h, t)$. For instance the triple $(Alice, Knows, Bob)$ corresponds to the first-order logic atom $Knows(Alice, Bob)$, which represents the fact that *Alice knows Bob*. Here, *Alice* and *Bob* are objects and *Knows* is a relation.

**Example 1.** *Let us have the following objects and relations:* $\mathcal{O} = \{France, Germany, Berlin, Paris\}$, $\mathcal{R} = \{CapitalOf, Borders\}$. *Then the knowledge graph, representing our geographical knowledge about these countries and cities, can look as* $\mathcal{G} = \{(France, Borders, Germany), (Germany, Borders, France), (Berlin, CapitalOf, Germany), (Paris, CapitalOf, France)\}$. *Whereas, in this case, the knowledge graph is complete and accurate, in reality most knowledge graphs will lack some of the true tuples. The task of knowledge base completion is to add such missing facts.*

### 2.2 KNOWLEDGE GRAPH EMBEDDING

Knowledge graph embedding methods represent knowledge graphs geometrically. They do so by representing objects and relations as vectors (or matrices). Given a knowledge graph $\mathcal{G}$ on some sets of objects $\mathcal{O}$ and relations $\mathcal{R}$, most knowledge graph embedding methods need three components: vector representations of objects, vector representations of relations and a function $\psi(\mathbf{x}_h, \mathbf{x}_r, \mathbf{x}_t)$ which takes the representations of $h$, $r$ and $t$ and produces a score that is used to assess how likely it is that the relation $(h, r, t)$ is contained in the knowledge graph. One can then compare the output of the function with a threshold to decide which triples are (predicted to be) in the knowledge graph.

**Note on notation.** For a set of objects $\mathcal{O}$ and a set of relations $\mathcal{R}$ we will normally denote their vector representations by $\mathbb{X}$. We will usually treat $\mathbb{X}$ as a look-up table. For an object $o$ we will typically use $\mathbf{x}_o$ to denote its vector representation from $\mathbb{X}$ and analogically for a relation $r$ we will use $\mathbf{x}_r$, unless explicitly stated otherwise. With a slight abuse of notation, we will also sometimes treat $\mathbb{X}$ as a set of vectors and, for instance, write $\mathbf{x} \in \mathbb{X}$. Relations can be mapped to matrices (denoted $\mathbf{M}_r$) or tensors instead of only vectors, and even if relations are mapped to vectors the dimension may be different from the dimension of entity embeddings. However, for the sake of simplicity, we will not introduce more notations, and $\mathbb{X}$ and $\psi$ will also be used in these cases.

### 2.3 EXISTING METHODS

One can roughly categorize knowledge graph (KG) embedding approaches into several groups based on different scoring functions such as distance-based scoring functions and similarity-based scoring functions.

The most well-known class of embedding methods using distance-based scoring functions is TransE (Bordes et al., 2013) and its variants. The scoring function of TransE is defined as the negative distance between $\mathbf{x}_h + \mathbf{x}_r$ and $\mathbf{x}_t$, i.e., $\psi(\mathbf{x}_h, \mathbf{x}_r, \mathbf{x}_t) = -||\mathbf{x}_h + \mathbf{x}_r - \mathbf{x}_t||_2$.

In order to effectively model more general relations rather than only one-to-one relations, TransH has been proposed (Wang et al., 2014). Given a triple $(h, r, t)$, the scoring function is defined as $-||\mathbf{x}_h^{\perp} + \mathbf{x}_r - \mathbf{x}_t^{\perp}||_2$ where $\mathbf{x}_h^{\perp}$ and $\mathbf{x}_t^{\perp}$ are the projections of entity embeddings $\mathbf{x}_h$ and $\mathbf{x}_t$ onto some hyperplane, respectively. TransR (Lin et al., 2015), TransD (Ji et al., 2015) and TranSparse (Ji et al., 2016) are similar to TransH, allowing relation-specific embeddings in different ways. TransM (Fan et al., 2014; Xiao et al., 2015), ManifoldE (Xiao et al., 2016) and TransF (Feng et al., 2016) relax the requirement $\mathbf{x}_h + \mathbf{x}_r \approx \mathbf{x}_t$.

SE (Bordes et al., 2011) uses two matrices to project head and tail entities for each relation $r$, and the score is $-||\mathbf{M}_r^1 \mathbf{x}_h - \mathbf{M}_r^2 \mathbf{x}_t||_1$.

RotatE (Sun et al., 2019) defines each relation as a rotation from the source entity to the target entity in the complex vector space, and its scoring function is $-||\mathbf{x}_h \circ \mathbf{x}_r - \mathbf{x}_t||_2$ where $\circ$ is the entry-wise product.

In the class of similarity-based methods, RESCAL (Nickel et al., 2011) and its variants are among the most well-known ones. RESCAL also assigns every entity a vector but every relation $r$ a matrix $\mathbf{M}_r$[1]. The scoring function is defined as $\mathbf{x}_h^{\top} \mathbf{M}_r \mathbf{x}_t$. Jenatton et al. (2012) further assumes that every $\mathbf{M}_r$ has the same set of eigenvectors. García-Durán et al. (2014) adds other terms into the scoring function $\mathbf{x}_h^{\top} \mathbf{M}_r \mathbf{x}_t + \mathbf{x}_h^{\top} \mathbf{x}_r + \mathbf{x}_r^{\top} \mathbf{x}_t + \mathbf{x}_t^{\top} D \mathbf{x}_h$ where $D$ is a diagonal matrix independent of relations. DistMult (Yang et al., 2015) is a simplified version of RESCAL by restricting $\mathbf{M}_r$ being diagonal, and hence improves the efficiency of RESCAL. To combine the advantages of both methods, i.e., the expressive power of RESCAL and the simplicity of DistMult, HolE (Nickel et al., 2016) has been proposed. ComplEx (Yang et al., 2015) extends DistMult by allowing complex-valued embeddings, which is more powerful in modeling asymmetric relations. Hayashi & Shimbo (2017) show that ComplEx is in fact equivalent to HolE. To capture analogical structures, ANALOGY (Liu et al., 2017) further requires $\mathbf{M}_r$ in RESCAL to be normal and mutually commutative.

A class of KG embeddings is based on tensor factorization approaches such as the Canonical Polyadic (CP) decomposition (Hitchcock, 1927) whose scoring function is similar to DistMult, but every entity has two independent embeddings. SimplE (Kazemi & Poole, 2018) is a recent extension of CP decomposition, which ensures the two embeddings of every entity dependent.

There are also a group of algorithms using neural networks for embedding, e.g., SME (Bordes et al., 2014), NTN and SLM (Bordes et al., 2014), MLP (Dong et al., 2014), NAM (Liu et al., 2016) and ConvE (Dettmers et al., 2018).

All the aforementioned embedding methods map entities and relations in a deterministic way. In KG2E (He et al., 2015), representations of entities and relations are random vectors sampled from multivariate normal distributions,

$$\mathbf{x}_h \sim \mathcal{N}(\boldsymbol{\mu}_h, \boldsymbol{\Sigma}_h) \quad \mathbf{x}_r \sim \mathcal{N}(\boldsymbol{\mu}_r, \boldsymbol{\Sigma}_r) \quad \mathbf{x}_t \sim \mathcal{N}(\boldsymbol{\mu}_t, \boldsymbol{\Sigma}_t).$$

A possible scoring function is $KL(\mathbf{x}_t - \mathbf{x}_h || \mathbf{x}_r)$ and another is the probability inner product (Jebara et al., 2004) of $\mathbf{x}_t - \mathbf{x}_h$ and $\mathbf{x}_r$. TransG (Han et al., 2016) further models relations using mixtures of Gaussians.

## 3 LEARNING SETTING

We work within the following learning setting. We assume that there exists some "ground truth" knowledge graph $\mathcal{G}^*$ and that we observe only a subset of the triples contained in it. We will denote the observed knowledge graph consisting of a subset of $\mathcal{G}^*$'s triples by $\widehat{\mathcal{G}}$. Furthermore, we assume that $\widehat{\mathcal{G}}$ is obtained from $\mathcal{G}^*$ by the following sampling process, called *data-generating distribution*.

**Definition 1** (Data-generating distribution). *Let $\mathcal{G}^*$ be a ground-truth knowledge graph and $\delta > 0$ be a real number. The data-generating distribution is given by the following process. Iterate over*

---

[1] One can also treat the embedding of every relation as a vector $\mathbf{x}_r$, and the corresponding scoring function becomes $\mathbf{x}_r \cdot vec(\mathbf{x}_h \mathbf{x}_t^{\top})$ where $vec$ is the vectorization operator.

*all triples $\tau \in \mathcal{G}^*$ and add each of them to $\widehat{\mathcal{G}}$ with probability $1 - \delta$ independently of the others. We call $\delta$ the* missingness parameter.

The task that we want to solve is then to reconstruct the ground-truth knowledge graph $\mathcal{G}^*$ as well as possible from a single sample from the data-generating distribution.

**Remark 1.** *Some knowledge graph embedding methods use different representations for the same object depending on whether it is used as a head or tail of a particular triple. It is easy to see that our general setting also allows such representations; it is enough to double the dimension of the vectors and assume that the function $\psi(\mathbf{x}_h, \mathbf{x}_r, \mathbf{x}_t)$ depends only on the first half of dimensions of $\mathbf{x}_h$ and the second half of the dimensions of $\mathbf{x}_t$. It is easy to see that we can also handle the case when the vector representations of relations have different dimensions than those of objects similarly.*

## 4  INTUITION AND SKETCH OF THE ARGUMENT

Our main results are bounds for the expected number of incorrectly predicted triples for knowledge graph completion. In particular, we show that, assuming certain technical assumption on the knowledge graph embedding method and the way it is trained, there exists a constant $C$ such that the expected error rate in the realizable setting is bounded by $C(\sqrt{|\mathcal{G}^*|/(|\mathcal{R}||\mathcal{O}|^2)} \cdot \sqrt{\ln |\mathcal{O}|/|\mathcal{O}|} + 1/|\mathcal{O}|)$, where $\mathcal{G}^*$ is the ground truth knowledge graph and $\mathcal{O}$ is the set of objects and $\mathcal{R}$ is the set of relations. The first term in the product measures sparsity of the graph and the second term has the familiar form of classical error bounds for learning from i.i.d. data. We also provide similar bounds for the agnostic setting.

The strategy taken in this paper is based on viewing the problem as learning a distribution on knowledge graphs. Suppose that we have not only one sample $\widehat{\mathcal{G}}$ (one knowledge graph) from the data-generating distribution, but actually many such samples $\widehat{\mathcal{G}}_1, \widehat{\mathcal{G}}_2, \ldots, \widehat{\mathcal{G}}_n$ (a collection of knowledge graphs instead of just one). We could try to use the knowledge graphs $\widehat{\mathcal{G}}_1, \widehat{\mathcal{G}}_2, \ldots, \widehat{\mathcal{G}}_n$ to estimate a distribution over knowledge graphs. Then we would predict the triples with frequency higher than some threshold $t$ as true. In reality, we do not have a collection of knowledge graphs but only one. However, to estimate the data-generating distribution, we can exploit the fact that we know its form and the independencies it satisfies. In the next section, we define a distribution to approximate the data-generating distribution, parameterized by the vector representations $\mathbb{X}$ of objects and relations.

For the approach sketched above to work, we need (i) the distribution to be expressive enough and (ii) we need to show that the learned distribution will be close enough to the data-generating distribution if we estimate it from a sufficiently large knowledge graph. The question of expressiveness has been tackled in the literature (e.g. Kazemi & Poole (2018)) and we do not deal with it much in this paper. We deal with the second question. Using a concentration inequality for log-likelihood (Theorem 1), we prove a generalization bound for log-likelihood (Theorem 2) from which a bound on Kullback-Leibler divergence follows. What then remains to be done is to show how to bound the number of incorrectly predicted triples as a function of the Kullback-Leibler divergence, which we do with the help of Pinsker's inequality (Theorem 3). Combining these two latter results together then yields a bound on the expected number of triples that are predicted incorrectly (both false positives and false negatives) by the learning method sketched here (Theorem 4).

## 5  A PROBABILISTIC PERSPECTIVE

Following the intuition described in the previous section, we cast the knowledge graph embedding problem as the problem of estimating the distribution induced by the data-generating process. We seek an approximation of this distribution as an exponential-family model of the form given below in Definition 2. The parameters of this exponential family model are the vector embeddings of the objects and relations.

**Definition 2** (KG-distribution). *Let $\mathcal{O}$ be a set of objects, $\mathcal{R}$ be a set of relations and $\Omega = 2^{\mathcal{O} \times \mathcal{R} \times \mathcal{O}}$ be the set of all possible knowledge graphs on these objects and relations. Let $\mathcal{G} \in \Omega$ be a knowledge graph and $\mathbb{X}$ be the vector representation of the objects and relations. Then we define*

$$P_{\mathbb{X}}(\mathcal{G}) = \frac{1}{Z} \prod_{(h,r,t) \in \mathcal{G}} \exp\left(\psi(\mathbf{x}_h, \mathbf{x}_r, \mathbf{x}_t)\right),$$

where $\mathbf{x}_h$, $\mathbf{x}_r$ and $\mathbf{x}_t$ are the vector representations of $h$, $r$ and $t$ given by $\mathbb{X}$, $\psi \colon \mathbb{R}^d \times \mathbb{R}^d \times \mathbb{R}^d$ is a function, $d$ is the dimension of the vector representations, and finally $Z = \sum_{\mathcal{G}' \in \Omega} \prod_{(h,r,t) \in \mathcal{G}'} \exp\left(\psi(\mathbf{x}_h, \mathbf{x}_r, \mathbf{x}_t)\right)$ is a normalization constant, ensuring that $P_{\mathbb{X}}(\mathcal{G})$ is a probability distribution.

Unless we allow $|\psi|$ to be infinite, KG-distributions cannot model the data-generating distribution exactly. However, as we will show it is enough if the KG-distributions are close enough in Kullback-Leibler divergence.

**Remark 2.** *The KG-distribution from Definition 2 can equivalently be written as*

$$P_{\mathbb{X}}(\mathcal{G}) = \prod_{(h,r,t) \in \mathcal{G}} \frac{\exp\left(\psi(\mathbf{x}_h, \mathbf{x}_r, \mathbf{x}_t)\right)}{1 + \exp\left(\psi(\mathbf{x}_h, \mathbf{x}_r, \mathbf{x}_t)\right)} \prod_{(h,r,t) \in (\mathcal{O} \times \mathcal{R} \times \mathcal{O}) \setminus \mathcal{G}} \frac{1}{1 + \exp\left(\psi(\mathbf{x}_h, \mathbf{x}_r, \mathbf{x}_t)\right)}.$$

## 6 GENERALIZATION BOUNDS FOR LOG-LIKELIHOOD

In this section, we derive generalization bounds for log-likelihood of KG-distributions learned from one sample (i.e. knowledge graph) from the data-generating distribution described in Section 3.

First, we define log-likelihood and normalized log-likelihood of the KG-distribution model.

**Definition 3** (Normalized log-likelihood). *Let $\mathcal{G}$ be a knowledge graph on a set of objects $\mathcal{O}$ and a set of relations $\mathcal{R}$ and $\mathbb{X}$ be vector representations of the objects and relations. The log-likelihood of this model is $L(\mathbb{X}|\mathcal{G}) = \ln P_{\mathbb{X}}(\mathcal{G})$ and its normalized log-likelihood is then defined as $NL(\mathbb{X}|\mathcal{G}) = \frac{1}{|\mathcal{O}|^2 |\mathcal{R}|} \cdot L(\mathbb{X}|\mathcal{G})$.*

We start by proving a concentration inequality for the log-likelihood of KG-distributions, which is a rather straightforward consequence of McDiarmid's inequality.

**Theorem 1.** *Let $\mathcal{G}^*$ be the ground-truth knowledge graph. Let $\mathcal{G}$ be sampled from the data-generating distribution induced by $\mathcal{G}^*$ and $\mathbb{X}$ be fixed vector representations. Then*

$$P\left[|L(\mathbb{X}|\mathcal{G}) - \mathbb{E}[L(\mathbb{X}|.)]| \geq \varepsilon\right] \leq 2 \exp\left(-\frac{2\varepsilon^2}{M^2 |\mathcal{G}^*|}\right)$$

*where $M \geq \sup_{\mathbf{h},\mathbf{t},\mathbf{r} \in \mathbb{X}} |\psi(\mathbf{x}_h, \mathbf{x}_r, \mathbf{x}_t)|$ (we assume $\sup_{\mathbf{h},\mathbf{t},\mathbf{r} \in \mathbb{X}} |\psi(\mathbf{x}_h, \mathbf{x}_r, \mathbf{x}_t)|$ to be finite) and the probability $P[\ldots]$ and the expectation $\mathbb{E}[\ldots]$ are w.r.t. the sampling of knowledge graphs from the data-generating distribution.*

The next theorem is the main result of this section. It bounds the expectation of the maximum deviation of the log-likelihood of KG distributions from their mean, where the KG-distributions are selected from a hypothesis class $\mathcal{H}$ (which has to satisfy certain assumptions listed in the theorem).

**Theorem 2.** *Let $\mathcal{G}^*$ be a ground-truth knowledge graph on a set of objects $\mathcal{O}$ and a set of relations $\mathcal{R}$ and let $\mathcal{G}$ be sampled by the data-generating distribution. Let $\mathcal{X} \subseteq \mathbb{R}^d$. Finally, let $\mathcal{H}_{\mathcal{X}}$ ("hypothesis class") denote the set of vector representations $\mathbb{X}$, $\mathbb{X} \subseteq \mathcal{X}^{|\mathcal{O}| + |\mathcal{R}|}$, of the objects from $\mathcal{O}$ and relations from $\mathcal{R}$. Let $D$, $M$ and $K$ be real numbers satisfying:*

1. *$\forall \mathbf{x}, \mathbf{x}' \in \mathcal{X}$: $\|\mathbf{x} - \mathbf{x}'\| \leq D$,*

2. *$\forall \mathbf{x}_h, \mathbf{x}_r, \mathbf{x}_t \in \mathcal{X}$: $|\psi(\mathbf{x}_h, \mathbf{x}_r, \mathbf{x}_t)| \leq M$,*

3. *$\forall \mathbf{x}_h, \mathbf{x}_r, \mathbf{x}_t, \mathbf{x}'_h, \mathbf{x}'_r, \mathbf{x}'_t \in \mathcal{X}$: $\|\psi(\mathbf{x}_h, \mathbf{x}_r, \mathbf{x}_t) - \psi(\mathbf{x}'_h, \mathbf{x}'_r, \mathbf{x}'_t)\| \leq K(\|\mathbf{x}_h - \mathbf{x}'_h\| + \|\mathbf{x}_r - \mathbf{x}'_r\| + \|\mathbf{x}_t - \mathbf{x}'_t\|).$*

*Then the following holds:*

$$\mathbb{E}\left[\sup_{\mathbb{X} \in \mathcal{H}_{\mathcal{X}}} |NL(\mathbb{X}|\mathcal{G}) - \mathbb{E}[NL(\mathbb{X}|.)]|\right] \leq \frac{4}{|\mathcal{O}|} + M \sqrt{\frac{|\mathcal{G}^*|}{|\mathcal{R}||\mathcal{O}|^2}} \cdot \sqrt{\frac{d \ln\left(4eDK|\mathcal{O}|\sqrt{d}\right)}{|\mathcal{O}|}}$$

*where both expectations on the l.h.s. are w.r.t. sampling of knowledge graphs from the data-generating distribution.*

# 7 PINSKER'S INEQUALITY COMES INTO PLAY

In this section, we use Pinsker's inequality to show how the Kullback-Leibler divergence between the data-generating distribution, which generates the observed knowledge graph, and the learned distribution can be used to bound the number of *incorrectly predicted triples*. In particular, we assume that all triples with probability given by the learned distribution greater than a fixed threshold $t$ are predicted as positive (i.e. predicted to belong in the knowledge graph) and the rest as negative (i.e. predicted not to belong in the knowledge graph). An *incorrectly predicted triple* is then either a triple which is included in the ground-truth knowledge graph but we predict it as false (false negative) or a triple which is not included in the ground-truth knowledge graph but we predict it as true (false positive). In combination with the results from the previous section, this will allow us to bound the expected number of incorrectly predicted triples (this is explained in detail in the next section).

Theorem 3, which is the main result of this section, bounds the number of incorrectly predicted triples as a function of the Kullback-Leibler divergence of the data-generating and the learned distributions, of the threshold $t$ (used to decide which triple is predicted as positive) and the missingness parameter $\delta$ (the fraction of the triples missing from the ground-truth knowledge graph). The main idea of the proof of this theorem (which is given in the appendix) is as follows. Whenever there is an incorrectly predicted triple $\tau$, the difference between its probability w.r.t. the data-generating distribution and w.r.t. the learned distribution must be greater than some fixed $\Delta$ (the value of $\Delta$ depends on the threshold $t$ and the missingness parameter $\delta$), otherwise $\tau$ could not be incorrectly predicted. Pinsker's inequality (Pinsker, 1964), which relates total variation distance and Kullback-Leibler divergence, then allows us to obtain a lower bound on the Kullback-Leibler divergence of the data-generating and the learned marginal distributions corresponding to the incorrectly predicted triple $\tau$. Since triples are sampled independently of each other in both the data-generating and the learned distributions, the sum of the Kullback-Leibler divergences of the marginal distributions corresponding to all possible individual triples will equal the Kullback-Leibler divergence of the data-generating distribution and the learned distribution. This all together is what allows us to derive the bound stated in the next theorem.

**Theorem 3.** *Let $\mathcal{G}^*$ be the ground-truth graph on a set of objects $\mathcal{O}$ and relations $\mathcal{R}$. Let $P$ denote the data-generating distribution induced by $\mathcal{G}^*$ with the missingness parameter $\delta$ and let $Q$ denote the learned distribution. Let $t \in (0; 1 - \delta)$ be a threshold for predicting which triples belong to the knowledge graph. Finally, let $\mathcal{F}$ be the set of incorrectly predicted triples, i.e. the set of all triples $\tau \in \mathcal{O} \times \mathcal{R} \times \mathcal{O}$ such that either $\tau \in \mathcal{G}^*$ and the probability of $\tau$ given by the learned distribution $Q$ is smaller than $t$ (false negatives) or $\tau \notin \mathcal{G}^*$ and the probability of $\tau$ given by the learned distribution $Q$ is greater or equal to $t$ (false positives). Then the following holds for the cardinality of the set of incorrectly predicted triples $\mathcal{F}$:*

$$|\mathcal{F}| \leq \max \left\{ \frac{\ln 2}{2t^2}, \frac{\ln 2}{2(1 - \delta - t)^2} \right\} \cdot KL(P||Q).$$

Note that the bound from this theorem is minimized for the threshold $t := (1 - \delta)/2$. This suggests how to set the threshold if we knew the missingness parameter $\delta$.

# 8 PUTTING EVERYTHING SO FAR TOGETHER

Next we combine the results from the previous sections to obtain a bound on the expected number of incorrectly predicted triples for vector representations $\mathbb{X}$ learned by maximizing the likelihood of the respective KG-distribution parametrized by $\mathbb{X}$. Here we assume that the domain $\mathcal{X}$ of the vector representations is a subset of $\mathbb{R}^d$ of finite diameter and that there is some pre-fixed scoring function $\psi$.

**Theorem 4.** *Let $\mathcal{G}^*$ be a ground-truth knowledge graph on a set of objects $\mathcal{O}$ and a set of relations $\mathcal{R}$ and let $\mathcal{G}$ be sampled by the data-generating distribution. Let $\mathcal{X} \subseteq \mathbb{R}^d$. Let $\mathcal{H}_\mathcal{X}$ ("hypothesis class") denote the set of vector representations $\mathbb{X}$, $\mathbb{X} \subseteq \mathcal{X}^{|\mathcal{O}| + |\mathcal{R}|}$, of the objects from $\mathcal{O}$ and relations from $\mathcal{R}$. Let $K$, $M$, $D$ be as in Theorem 2 and $t$ and $\delta$ be as in Theorem 3. Let $\mathbb{X} \in \mathcal{H}_\mathcal{X}$ be a representation of objects and relations learned by maximizing the log-likelihood $L(\mathbb{X}|\mathcal{G})$, i.e. $\mathbb{X} = \arg\max_{\mathbb{X}' \in \mathcal{H}_\mathcal{X}} L(\mathbb{X}'|\mathcal{G})$. Finally, let $\mathbb{X}^* = \arg\max_{\mathbb{X} \in \mathcal{H}_\mathcal{X}} \mathbb{E}[L(\mathbb{X}|.)]$. Then the following holds for*

the sets of incorrectly predicted triples $\mathcal{F}_{\mathbb{X}}$, obtained using the KG-distribution with $\mathbb{X}$:

$$
\mathbb{E}\left[\frac{|\mathcal{F}_{\mathbb{X}}|}{|\mathcal{O}|^2|\mathcal{R}|}\right] \leq \max\left\{\frac{\ln 2}{t^2}, \frac{\ln 2}{(1-\delta-t)^2}\right\} \cdot \left(\frac{4}{|\mathcal{O}|}\right.
$$
$$
\left. +M\sqrt{\frac{|\mathcal{G}^*|}{|\mathcal{R}||\mathcal{O}|^2}} \cdot \sqrt{\frac{d\ln\left(4eDK|\mathcal{O}|\sqrt{d}\right)}{|\mathcal{O}|} + \frac{KL(P||P_{\mathbb{X}^*})}{|\mathcal{O}|^2|\mathcal{R}|}}\right)
$$

where $KL(P||P_{\mathbb{X}^*})$ is the Kullback-Leibler divergence of the data-generating distribution $P$ and the best possible representable distribution $P_{\mathbb{X}^*}$.

# 9 THE "REALIZABLE" CASE

In learning theory, one usually speaks of the "realizable case" when the hypothesis class contains the target concept. In general, the data generating distribution cannot be represented exactly using the KG-distribution with vector representations from a given set $\mathcal{X}$ unless we allow $|\psi|$ to be infinite (which would be needed to model the triples that have zero probability of being sampled from the data generating distribution because they are not contained in the ground-truth knowledge graph $\mathcal{G}^*$). Therefore we cannot really speak of the "realizable case" in the usual sense. In particular it may be the case that the given scoring function $\psi$ and the given domain of vector representations $\mathcal{X}$ are enough to represent the knowledge graph in the sense that there exists a threshold $t$ such that $(\psi(\mathbf{x}_h, \mathbf{x}_r, \mathbf{x}_t) \geq t) \Leftrightarrow (h, r, t) \in \mathcal{G}^*)$, and yet the data-generating distribution could not be represented by the corresponding KG-distribution exactly. Hence, it makes sense to define the realizable case slightly differently for our setting.

**Definition 4** ($\gamma$-Realizable case). *For a given data-generating distribution $P$ induced by a ground-truth knowledge graph $\mathcal{G}^*$ with the missingness parameter $\delta$ and a given scoring function $\psi$ and a domain of vector representations $\mathcal{X}$, we say that a KG-distribution $P_{\mathbb{X}}$ given by a scoring function $\psi$ and vector representations $\mathbb{X}$ is $\gamma$-admissible if the following is satisfied: $\exp(\psi(\mathbf{x}_h, \mathbf{x}_r, \mathbf{x}_t))/(1 + \exp(\psi(\mathbf{x}_h, \mathbf{x}_r, \mathbf{x}_t))) \geq 1 - \gamma$ for all $(h, r, t) \in \mathcal{G}^*$ and $1/(1 + \exp(\psi(\mathbf{x}_h, \mathbf{x}_r, \mathbf{x}_t))) \leq \gamma$ for all $(h, r, t) \notin \mathcal{G}^*$, where $\mathbf{x}_h, \mathbf{x}_r, \mathbf{x}_t$ are vector representations given by $\mathbb{X}$. We say that a learning problem is $\gamma$-realizable if all vector representations $\mathbb{X} \in \mathcal{X}^{|\mathcal{O}|+|\mathcal{R}|}$ that maximize the* **expected** *log-likelihood (i.e. the best vector representations) give rise to $\gamma$-admissible KG-distributions.*

Next we present a version of Theorem 3 suitable for $\gamma$-realizable settings.

**Theorem 5.** *Let $\mathcal{G}^*$ be the ground-truth graph on a set of objects $\mathcal{O}$ and relations $\mathcal{R}$. Let $P$ denote the data-generating distribution induced by $\mathcal{G}^*$ and let $P_\gamma$ be a $\gamma$-admissible distribution. Let $Q$ denote the learned distribution and let $t \in (\gamma; 1 - \gamma)$ be a threshold. Then the following holds for the cardinality of the set of incorrectly predicted triples $\mathcal{F}$:*

$$
|\mathcal{F}| \leq \max\left\{\frac{\ln 2}{2(t-\gamma)^2}, \frac{\ln 2}{2(1-\gamma-t)^2}\right\} \cdot KL(P_\gamma||Q).
$$

In the next theorem we present a tighter version of Theorem 4 for $\gamma$-realizable settings.

**Theorem 6.** *Let everything be as in Theorem 4 but assume a $\gamma$-realizable setting. Then the following holds for the sets of incorrectly predicted triples $\mathcal{F}_{\mathbb{X}}$, obtained using the KG-distribution with $\mathbb{X}$:*

$$
\mathbb{E}\left[\frac{|\mathcal{F}_{\mathbb{X}}|}{|\mathcal{O}|^2|\mathcal{R}|}\right] \leq \max\left\{\frac{\ln 2}{(t-\gamma)^2}, \frac{\ln 2}{(1-t-\gamma)^2}\right\} \cdot \left(\frac{4}{|\mathcal{O}|} + M\sqrt{\frac{|\mathcal{G}^*|}{|\mathcal{R}||\mathcal{O}|^2}}\sqrt{\frac{d\ln\left(4eDK|\mathcal{O}|\sqrt{d}\right)}{|\mathcal{O}|}}\right)
$$

The next example illustrates that the results from Theorem 6 are asymptotically non-trivial (which is non-obvious as we show below).

**Example 2.** *Let us assume that we have some fixed $\mathcal{H}_{\mathcal{X}}$, $K$, $D$ and $d$ and that the $\mathbb{X}^* \in \mathcal{H}_{\mathcal{X}}$. For simplicity, let us assume that this is a $\gamma$-realizable setting with $\gamma = 0.1$. Let the ground knowledge graphs $\mathcal{G}$ be sparse and satisfy $|\mathcal{G}| = \alpha(|\mathcal{O}|) \cdot |\mathcal{O}|$ where $\alpha$ is an integer-valued function and $\mathcal{O}$ is the set of objects in the knowledge graph. We suppose that the threshold $t$ satisfies $t \in (\gamma, 1 - \gamma)$. Then the expected fraction of triples incorrectly predicted by the learned model from our hypothesis*

*class can be bounded by: $C_1 \cdot \sqrt{\alpha(|\mathcal{O}|)}\sqrt{\ln|\mathcal{O}|}/|\mathcal{O}|$ where $C_1$ is a positive constant. Now consider another "alternative prediction method" which always predicts all triples as false. The fraction of triples incorrectly predicted by this method will be: $\mathbb{E}\left[|\mathcal{F}|/(|\mathcal{O}|^2|\mathcal{R}|)\right] \leq C_2 \cdot \alpha(|\mathcal{O}|)/|\mathcal{O}|$ where $C_2$ is a positive constant. It follows that whenever $\alpha(|\mathcal{O}|)$ grows faster than $\ln|\mathcal{O}|$, the bounds of the maximum likelihood method, derived in this paper, become tighter than the error bound of this trivial method. Here, we note that $\alpha(|\mathcal{O}|)$ growing only as a logarithm corresponds to very sparse knowledge graphs (in which prediction of positive examples is likely hard).*

## 10 EXISTING KGE METHODS IN THE LIGHT OF OUR ANALYSIS

In this section, we discuss how existing knowledge graph embedding methods in general fit into our analysis.

### 10.1 SCORING FUNCTIONS

We assumed that predictions are made by comparing $\exp\left(\psi(\mathbf{x}_h, \mathbf{x}_r, \mathbf{x}_t)\right)$ with a threshold. In existing works, one directly compares the output of the scoring function $\psi(\mathbf{x}_h, \mathbf{x}_r, \mathbf{x}_t)$ with a threshold. This differs only by taking a logarithm and, hence, obviously does not harm applicability of our analysis.

### 10.2 LOSS FUNCTIONS

Throughout this paper we assumed learning vector representations of objects and relations using maximum likelihood. Here we show that this setting corresponds to learning knowledge graph embeddings using a sigmoidal transformation of the scoring function $\psi$ together with log-loss. This combination is often used in the knowledge graph embedding literature (Trouillon et al., 2016; Dettmers et al., 2018; Kazemi & Poole, 2018).[2] In particular the loss used in the literature is (using our notation):

$$
\begin{aligned}
l(\mathbb{X}, \mathcal{G}) &= \sum_{(h,r,t)\in\mathcal{G}} \ln\left(1 + \exp(-\psi(\mathbf{x}_h, \mathbf{x}_r, \mathbf{x}_t))\right) + \sum_{(h,r,t)\in(\mathcal{O}\times\mathcal{R}\times\mathcal{O})\setminus\mathcal{G}} \ln\left(1 + \exp(\psi(\mathbf{x}_h, \mathbf{x}_r, \mathbf{x}_t))\right) \\
&= \sum_{(h,r,t)\in\mathcal{G}} -\ln\left(\frac{1}{1 + \exp(-\psi(\mathbf{x}_h, \mathbf{x}_r, \mathbf{x}_t))}\right) + \sum_{(h,r,t)\in(\mathcal{O}\times\mathcal{R}\times\mathcal{O})\setminus\mathcal{G}} -\ln\left(\frac{1}{1 + \exp(\psi(\mathbf{x}_h, \mathbf{x}_r, \mathbf{x}_t))}\right) \\
&= -\left(\sum_{(h,r,t)\in\mathcal{G}} \ln\left(\frac{\exp(\psi(\mathbf{x}_h, \mathbf{x}_r, \mathbf{x}_t))}{1 + \exp(\psi(\mathbf{x}_h, \mathbf{x}_r, \mathbf{x}_t))}\right) + \sum_{(h,r,t)\in(\mathcal{O}\times\mathcal{R}\times\mathcal{O})\setminus\mathcal{G}} \ln\left(\frac{1}{1 + \exp(\psi(\mathbf{x}_h, \mathbf{x}_r, \mathbf{x}_t))}\right)\right) .
\end{aligned}
$$

It is not difficult to see that this is the same as the negative log-likelihood of the KG-distribution with scoring function $\psi$. Thus, our analysis directly applies to methods that use this loss, such as ComplEx (Trouillon et al., 2016), ConvE (Dettmers et al., 2018) and SimplE (Kazemi & Poole, 2018). For other methods, log-loss can in principle be also used and it was observed by Trouillon & Nickel (2017) that the margin-based loss functions, used by many knowledge graph embedding methods, are more prone to overfitting compared to log-likelihood. Our results may shed further light on this latter observation, however, to verify it theoretically would require to perform a similar type of analysis as we did in this paper for the margin-based loss functions (which might require completely different techniques). Therefore we leave this comparison for future work.

### 10.3 LIPSCHITZ CONTINUITY AND BOUNDED DOMAIN ASSUMPTIONS

Our result requires Lipschitz continuity. Most of the scoring functions $\psi$ used by existing knowledge graph methods are not globally Lipschitz continuous. However, they are Lipschitz continuous if we

---

[2]These works use regularized log-loss. Here we ignore the regularization term but note that bounding the domain of the vector representations has to some extent a similar effect.

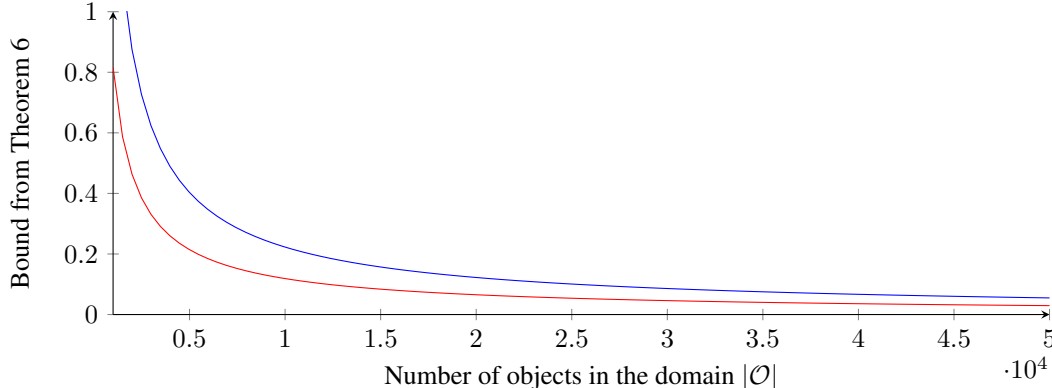

Figure 1: Illustration of the generalization bound from Theorem 6 applied on TransE (Bordes et al., 2013) and SimplE (Kazemi & Poole, 2018). The bound for TransE is shown in red and the bound for SimplE in blue. The parameters of the models are described in the main text.

restrict the domain of the embeddings to a bounded subset of $\mathbb{R}^d$ (this is among others true for the approaches based on matrix or tensor factorization).

For obtaining the generalization bounds, we needed to assume that the learned vector representations come from a set with a bounded diameter. Existing methods usually do not have such a constraint explicitly but use a regularization term which has a similar effect in practice.

### 10.4 NUMERICAL EXAMPLES

**TransE** Here we show how our results apply to TransE (Bordes et al., 2013), arguably the simplest embedding method. In TransE, $\psi(\mathbf{x}_h, \mathbf{x}_r, \mathbf{x}_t) = -\|\mathbf{x}_h + \mathbf{x}_r - \mathbf{x}_t\|$. It is not difficult to show that the Lipschitz constant $K = 1$ as follows (using the triangle inequality repeatedly):

$$\|\|\mathbf{x}_h + \mathbf{x}_r - \mathbf{x}_t\| - \|\mathbf{x}'_h + \mathbf{x}'_r - \mathbf{x}'_t\|\| = \|\|\mathbf{x}_h - \mathbf{x}'_h + \mathbf{x}_r - \mathbf{x}'_r - \mathbf{x}_t + \mathbf{x}'_t + \mathbf{x}'_h + \mathbf{x}'_r - \mathbf{x}'_t\|$$
$$-\|\mathbf{x}'_h + \mathbf{x}'_r - \mathbf{x}'_t\|\| \leq \|\mathbf{x}_h - \mathbf{x}'_h + \mathbf{x}_r - \mathbf{x}'_r - \mathbf{x}_t + \mathbf{x}'_t\| \leq \|\mathbf{x}_h - \mathbf{x}'_h\| + \|\mathbf{x}_r - \mathbf{x}'_r\| + \|\mathbf{x}_t - \mathbf{x}'_t\|.$$

Let the dimension of the vector embeddings be $d = 3$. Let us suppose that $\mathcal{X} = \{\mathbf{x} | \|\mathbf{x}\| \leq \ln 10\}$. Then the remaining parameters from Theorem 4 are $D = \ln 10^2$ and $M = \ln 10^3$. For simplicity, let us assume a $\gamma$-realizable setting with $\gamma = 0.01$. Let $t = 0.5$. Finally suppose that $|\mathcal{G}^*| = |\mathcal{O}| \ln^2 |\mathcal{O}|$ and $|\mathcal{R}| = 1$. Now we have all parameters needed to apply Theorem 6. We plot the resulting bound for a varying number of objects in Figure 1 in red. Note that, even though the number of dimensions of the vector embeddings is low, the total number of parameters of the model is very high because each object and each relation has its own parameters.

**SimplE** Next we illustrate how our analysis applies to SimplE (Kazemi & Poole, 2018), which is a fully-expressive knowledge graph embedding method. In SimplE, each object $o \in \mathcal{O}$ and each relation $r \in \mathcal{R}$ are represented by two vectors $\mathbf{x}'_o$ and $\mathbf{x}''_o$, respectively, $\mathbf{x}'_r$ and $\mathbf{x}''_r$. In our formalism, we represent both as single vectors: $\mathbf{x}_o = [\mathbf{x}'_o, \mathbf{x}''_o]$ and $\mathbf{x}_r = [\mathbf{x}'_r, \mathbf{x}''_r]$. The scoring function of SimplE is $\psi(\mathbf{x}_h, \mathbf{x}_r, \mathbf{x}_t) = \frac{1}{2}(\langle \mathbf{x}'_h, \mathbf{x}'_r, \mathbf{x}''_t \rangle + \langle \mathbf{x}'_t, \mathbf{x}''_r, \mathbf{x}''_h \rangle)$ where $\langle \mathbf{x}, \mathbf{y}, \mathbf{z} \rangle := \sum_{i=1}^d x_i y_i z_i$. Clearly the Lipschitz constant of $\psi(\mathbf{x}_o, \mathbf{x}_r, \mathbf{x}_t)$ is not bounded for unbounded $\mathbf{x}_h$, $\mathbf{x}_r$ and $\mathbf{x}_t$. We therefore need to restrict the domain of the vector representations $\mathcal{X}$, which our theorems assume anyways and which is done implicitly through the L2-regularization term in the original work (Kazemi & Poole, 2018). We show in the appendix that the Lipschitz constant $K$ for SimplE can be bounded by $\frac{\sqrt{6}}{2} \sup_{\mathbf{x} \in \mathcal{X}} \|\mathbf{x}\|^2$. For the illustration, we will assume the same dimension $d = 3$ and the same domain $\mathcal{X}$ as we did for TransE, i.e. $\mathcal{X} = \{\mathbf{x} | \|\mathbf{x}\| \leq \ln 10\}$. Then the other parameters needed to compute the generalization bound for SimplE are $D = \ln 10^2$, $M = (\ln 10)^3$, $K = \frac{\sqrt{6}}{2}(\ln 10)^2$. Finally, we assume the same $\gamma$, $|\mathcal{R}|$ and $|\mathcal{G}^*|$ as we did for TransE. The resulting generalization bound for SimplE following from Theorem 5 is shown in Figure 1 in red together with the bound for TransE (in blue). Not surprisingly, the generalization error bound is looser for SimplE as it is

a more flexible model (in fact, a fully-expressive model with a sufficiently large dimension of the vector representations).

## 11 RELATED WORK

Our work is very close in spirit to the theoretical analyses of learning from positive and unlabeled examples, known as PU-learning (Niu et al., 2016). However, there are several distinctions that do not allow direct application of the existing results to the problem that we analyze in this paper. First, existing works on PU-learning assume the inductive setting whereas our setting is transductive. Second, the work (Niu et al., 2016), which is one of the few papers analyzing generalization bounds for PU-learning, only applies to losses that satisfy $l(t, +1) + l(t, -1) = 1$ which is not the case for log-loss. Moreover the losses considered by Niu et al. (2016) are usually not used for knowledge graph embedding in practice. Third, even if the previous two differences were not a problem (although they are), it would still not be obvious how to apply the existing results to the knowledge graph embedding setting in which we deal with interconnected examples; i.e. it is not clear what the attributes of the examples and the hypothesis class should be so that we could fit knowledge graph embedding setting into the standard PU learning setting. Lastly, an advantage of our analysis is that it uses a simple probabilistic model which provides a possible, hopefully intuitive, explanation of why knowledge graph embedding methods work (at least in the standard testing scenarios that use test data generated completely at random).

Another line of works that are related to ours are works on generalization bounds for link prediction, such as (London et al., 2013; Wang et al., 2018). The main differences w.r.t. these works is that (i) they assume availability of both positive and negative examples (whereas we only have positive examples), (ii) the links are predicted based on attributes of nodes and not on the structure of the relations. Finally, our work is also related to works for tensor completion using tensor decomposition, such as (Nickel & Tresp, 2013). However, first, not all knowledge graph embedding methods can be seen as some low-rank tensor decompositions and, second, as far as we know no analysis of generalization bounds of tensor decomposition applies to the PU-learning setting, in which one observes only positive examples (which is the setting studied in our work).

## 12 CONCLUSIONS

In this paper we have derived generalization bounds for the expected number of triples predicted incorrectly by knowledge graph embedding methods. The main assumptions that we used were: (i) facts are missing from the knowledge graph "completely at random" and (ii) the vector representations of objects and relations are learned by maximizing log-likelihood of the model (equivalently by minimizing log-loss). While the "missing completely at random" assumption may seem simplistic, it is actually how training and test sets are usually constructed, e.g. in (Bordes et al., 2013), and only recently there have been works systematically looking beyond datasets constructed in this way, e.g. (Trouillon et al., 2019). As for the second assumption, it would be interesting to see if one could derive similar bounds for other loss functions that have been used for knowledge graph embedding in the literature, which would then provide a more solid theoretical justification for their use.

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

## A USEFUL INEQUALITIES

We will need the following well-known inequalities.

**Lemma 7** (Pinsker's inequality (Pinsker, 1964))**.** *Let $P$ and $Q$ be distributions. Then*

$$KL(P\|Q) \geq \frac{1}{2\ln 2}\|P - Q\|_1^2.$$

**Lemma 8** (McDiarmid's Inequality (McDiarmid, 1989))**.** *Let $X_1, \ldots, X_n$ be independent random variables with values from a set $\mathcal{X}$ and let $f : \mathcal{X}^n \to \mathbb{R}$. If, for all $i \in \{1, \ldots, n\}$, and for all $x_1, , x_n, x_i' \in \mathcal{X}$, the function $f$ satisfies $|f(x_1, \ldots, x_i, \ldots, x_n) - f(x_1, \ldots, x_i', \ldots, x_n)| \leq c_i$ then:*

$$P[|f(X_1, \ldots, X_n) - \mathbb{E}[f]| \geq \varepsilon] \leq 2\exp\left(-\frac{2\varepsilon^2}{\sum_{i=1}^n c_i^2}\right).$$

## B PROOF OF THEOREM 1

In this section, we prove Theorem 1.

**Theorem 1.** *Let $\mathcal{G}^*$ be the ground-truth knowledge graph. Let $\mathcal{G}$ be sampled from the data-generating distribution induced by $\mathcal{G}^*$ and $\mathbb{X}$ be fixed vector representations. Then*

$$P\left[|L(\mathbb{X}|\mathcal{G}) - \mathbb{E}[L(\mathbb{X}|.)]| \geq \varepsilon\right] \leq 2\exp\left(-\frac{2\varepsilon^2}{M^2|\mathcal{G}^*|}\right),$$

*where $M \geq \sup_{\mathbf{h},\mathbf{t}\in\mathbb{X}_o,\mathbf{r}\in\mathbb{X}_r} |\psi(\mathbf{x}_h, \mathbf{x}_r, \mathbf{x}_t)|$.*

*Proof.* Let $F := |\mathcal{G}^*|$. Let $U_1, \ldots, U_F$ be Bernoulli random variables such that $P[U_i = 1] = 1 - \delta$. Let us denote by $\tau_i$ the $i$-th tuple in $\mathcal{G}^*$. Let $g : \{0,1\}^F \to 2^{\mathcal{G}^*}$ be defined as $g(u_1, \ldots, u_F) = \{\tau_i \in \mathcal{G}^*|u_i = 1\}$, i.e. $g(u_1, \ldots, u_F)$ contains all $\tau_i$'s for which $u_i = 1$. It is not difficult to see that $\mathcal{G}$ and $g(U_1, \ldots, U_F)$ have the same distribution. Hence we can bound the probability $P\left[|L(\mathbb{X}|g(U_1, \ldots, U_F)) - \mathbb{E}[L(\mathbb{X}|.)]| \geq \varepsilon\right]$ instead of directly $P\left[|L(\mathbb{X}|\mathcal{G}) - \mathbb{E}[L(\mathbb{X}|.)]| \geq \varepsilon\right]$.

It follows from the assumptions of the theorem that for any $(u_1, \ldots, u_F) \in \{0,1\}^F$ the following must hold:

$$|L(\mathbb{X}|g(u_1, \ldots, u_i, \ldots, u_F)) - L(\mathbb{X}|g(u_1, \ldots, u_i', \ldots, u_F))|$$

$$= \left|\sum_{(h,r,t)\in g(u_1,\ldots,u_i,\ldots,u_F)} \psi(\mathbf{x}_h, \mathbf{x}_r, \mathbf{x}_t) - \ln Z - \sum_{(h,r,t)\in g(u_1,\ldots,u_i',\ldots,u_F)} \psi(\mathbf{x}_h, \mathbf{x}_r, \mathbf{x}_t) + \ln Z\right|$$

$$\leq |\psi(\mathbf{x}_{h_i}, \mathbf{x}_{r_i}, \mathbf{x}_{t_i})| \leq M,$$

where $\mathbf{x}_h$, $\mathbf{x}_r$ and $\mathbf{x}_t$ denote vector representations of $h$, $r$ and $t$, respectively.

Hence, we can use McDiarmid's inequality to obtain $P\left[|L(\mathbb{X}|g(U_1, \ldots, U_F)) - \mathbb{E}[L(\mathbb{X}|.)]| \geq \varepsilon\right] \leq 2\exp\left(-\frac{2\varepsilon^2}{M^2 F}\right)$ from which the statement of the theorem follows directly. $\square$

## C PROOF OF THEOREM 2

In this section, we give a proof of Theorem 2. First, we prove several lemmas that we need to prove this theorem.

**Lemma 9.** *Let $\widetilde{M} > 0$ be such that for every object or relation $o$ it holds $\|\mathbf{x}_o - \mathbf{x}_o'\| \leq \widetilde{M}$ where $\mathbf{x}_o$ and $\mathbf{x}_o'$ are the vector representations of $o$ in $\mathbb{X}$ and $\mathbb{X}'$, respectively. If $\psi$ satisfies the assumptions of Theorem 2 then the following holds: $|L(\mathbb{X}_o, \mathbb{X}_r|\mathcal{G}) - L(\mathbb{X}_o', \mathbb{X}_r'|\mathcal{G})| \leq 2K|\mathcal{O}|^2|\mathcal{R}|\widetilde{M}$.*

*Proof.* Let us first introduce some notation. Let $V := |\mathcal{O}|$ and $R = |\mathcal{R}|$. Whenever we write $\mathbf{x}_h$, $\mathbf{x}_r$ and $\mathbf{x}_t$, it denotes the vector representations of $h$, $r$ and $t$ from $\mathbb{X}$, whereas when we write $\mathbf{x}'_h$, $\mathbf{x}'_r$ and $\mathbf{x}'_t$, it denotes the representations from $\mathbb{X}'$. We have

$$|L(\mathbb{X}_o, \mathbb{X}_r|\mathcal{G}) - L(\mathbb{X}'_o, \mathbb{X}'_r|\mathcal{G})| = \left| \sum_{(h,r,t)\in\mathcal{G}} (\psi(\mathbf{x}_h, \mathbf{x}_r, \mathbf{x}_t) - \psi(\mathbf{x}'_h, \mathbf{x}'_r, \mathbf{x}'_t)) - \ln\frac{Z}{Z'} \right|$$

$$\leq \left| \sum_{(h,r,t)\in\mathcal{G}} (\psi(\mathbf{x}_h, \mathbf{x}_r, \mathbf{x}_t) - \psi(\mathbf{x}'_h, \mathbf{x}'_r, \mathbf{x}'_t)) \right| + \left| \ln\frac{Z}{Z'} \right|.$$

Next we bound the terms on the r.h.s. of the above inequality separately. We start by obtaining the bound for the first term:

$$\left| \sum_{(h,r,t)\in\mathcal{G}} (\psi(\mathbf{x}_h, \mathbf{x}_r, \mathbf{x}_t) - \psi(\mathbf{x}'_h, \mathbf{x}'_r, \mathbf{x}'_t)) \right|$$

$$\leq KV^2R \cdot (\|\mathbf{x}_h - \mathbf{x}'_h\| + \|\mathbf{x}_r - \mathbf{x}'_r\| + \|\mathbf{x}_t - \mathbf{x}'_t\|) \leq KV^2R\widetilde{M}. \quad (1)$$

It remains to bound the second term. We have:

$$\frac{Z}{Z'} = \frac{\sum_{\mathcal{G}\in\Omega}\prod_{(h,r,t)\in\mathcal{G}} e^{\psi(\mathbf{x}_h,\mathbf{x}_r,\mathbf{x}_t)}}{\sum_{\mathcal{G}\in\Omega}\prod_{(h,r,t)\in\mathcal{G}} e^{\psi(\mathbf{x}'_h,\mathbf{x}'_r,\mathbf{x}'_t)}} = \frac{\sum_{\mathcal{G}\in\Omega}\prod_{(h,r,t)\in\mathcal{G}} e^{(\psi(\mathbf{x}_h,\mathbf{x}_r,\mathbf{x}_t) - \psi(\mathbf{x}'_h,\mathbf{x}'_r,\mathbf{x}'_t) + \psi(\mathbf{x}'_h,\mathbf{x}'_r,\mathbf{x}'_t))}}{\sum_{\mathcal{G}\in\Omega}\prod_{(h,r,t)\in\mathcal{G}} e^{\psi(\mathbf{x}'_h,\mathbf{x}'_r,\mathbf{x}'_t)}}$$

$$\leq e^{K \cdot V^2 R \cdot \widetilde{M}} \frac{\sum_{\mathcal{G}\in\Omega}\prod_{(h,r,t)\in\mathcal{G}} e^{\psi(\mathbf{x}'_h,\mathbf{x}'_r,\mathbf{x}'_t)}}{\sum_{\mathcal{G}\in\Omega}\prod_{(h,r,t)\in\mathcal{G}} e^{\psi(\mathbf{x}'_h,\mathbf{x}'_r,\mathbf{x}'_t)}} = e^{KV^2R\widetilde{M}} \quad (2)$$

where the last inequality follows from the inequality $K(\|\mathbf{x}_h - \mathbf{x}'_h\| + \|\mathbf{x}_r - \mathbf{x}'_r\| + |\mathbf{x}_t - \mathbf{x}'_t\|) \leq K\widetilde{M}$. Combining (1) and (2) we obtain the inequality that we needed to prove. □

**Lemma 10.** Let $\mathcal{G}^*$ be a ground-truth knowledge graph on a set of objects $\mathcal{O}$ and a set of relations $\mathcal{R}$ and let $\mathcal{G}$ be sampled by the data-generating distribution. Let $\mathcal{H}$ ("hypothesis class") be a **finite** set of vector representations of the objects from $\mathcal{O}$ and relations from $\mathcal{R}$. Then the following inequality holds

$$\mathbb{E}\left[ \sup_{\mathbb{X}\in\mathcal{H}} |L(\mathbb{X}|\mathcal{G}) - \mathbb{E}[L(\mathbb{X}|.)| \right] \leq M\sqrt{\frac{|\mathcal{G}^*|\ln(2e|\mathcal{H}|)}{2}}$$

where $M \geq \sup_{\mathbf{h},\mathbf{t}\in\mathbb{X}_o, \mathbf{r}\in\mathbb{X}_r} |\psi(\mathbf{x}_h, \mathbf{x}_r, \mathbf{x}_t)|$.

*Proof.* We denote $W := \sup_{\mathbb{X}\in\mathcal{H}} |L(\mathbb{X}|\mathcal{G}) - \mathbb{E}[L(\mathbb{X}|.)|$ and $B := 2/(|\mathcal{G}^*| \cdot M^2)$. Then, using the union bound and Theorem 1, we have: $P[W \geq \varepsilon] \leq 2|\mathcal{H}| \cdot \exp(-\varepsilon^2 \cdot B)$. Next, we have

$$\mathbb{E}[W] \leq \sqrt{\mathbb{E}[W^2]} = \sqrt{\int_0^\infty P[W^2 \geq t]dt} = \sqrt{\int_0^\infty P[W \geq \sqrt{t}]dt} \leq \sqrt{u + \int_u^\infty P[W \geq \sqrt{t}]dt}$$

$$\leq \sqrt{u + \int_u^\infty 2|\mathcal{H}| \cdot \exp(-t \cdot B)dt} = \sqrt{u + \frac{2|\mathcal{H}|}{B} \cdot \exp(-u \cdot B)}$$

The above expression is minimized for $u := \ln(2|\mathcal{H}|)/B$ (note that $u$ may be arbitrary), yielding the bound:

$$\sqrt{\frac{\ln(2e|\mathcal{H}|)}{B}} = M\sqrt{\frac{|\mathcal{G}^*|\ln(2e|\mathcal{H}|)}{2}}.$$

□

**Lemma 11** (Covering number, e.g., (Shalev-Shwartz & Ben-David, 2014)). *Let $\mathcal{S}$ be a subset of $\mathbb{R}^d$ of diameter at most $k$ (i.e. for any $\mathbf{x}_1, \mathbf{x}_2 \in \mathcal{S}$, $\|\mathbf{x}_1 - \mathbf{x}_2\| \leq k$). Then $\mathcal{S}$ can be covered by $\left((2k\sqrt{d})/\rho\right)^d$ balls of radius $\rho$.*

Now we are ready to prove Theorem 2.

**Theorem 2.** *Let $\mathcal{G}^*$ be a ground-truth knowledge graph on a set of objects $\mathcal{O}$ and a set of relations $\mathcal{R}$ and let $\mathcal{G}$ be sampled by the data-generating distribution . Let $\mathcal{X} \subseteq \mathbb{R}^d$. Finally, let $\mathcal{H}_{\mathcal{X}}$ ("hypothesis class") denote the set of vector representations $\mathbb{X}$, $\mathbb{X} \subseteq \mathcal{X}^{|\mathcal{O}|+|\mathcal{R}|}$, of the objects from $\mathcal{O}$ and relations from $\mathcal{R}$. Let $D$, $M$ and $K$ be real numbers satisfying:*

1. $\forall \mathbf{x}, \mathbf{x}' \in \mathcal{X}$: $\|\mathbf{x} - \mathbf{x}'\| \leq D$,

2. $\forall \mathbf{x}_h, \mathbf{x}_r, \mathbf{x}_t \in \mathcal{X}$: $|\psi(\mathbf{x}_h, \mathbf{x}_r, \mathbf{x}_t)| \leq M$,

3. $\forall \mathbf{x}_h, \mathbf{x}_r, \mathbf{x}_t, \mathbf{x}'_h, \mathbf{x}'_r, \mathbf{x}'_t \in \mathcal{X}$: $\|\psi(\mathbf{x}_h, \mathbf{x}_r, \mathbf{x}_t) - \psi(\mathbf{x}'_h, \mathbf{x}'_r, \mathbf{x}'_t)\| \leq K(\|\mathbf{x}_h - \mathbf{x}'_h\| + \|\mathbf{x}_r - \mathbf{x}'_r\| + \|\mathbf{x}_t - \mathbf{x}'_t\|)$,

*Then the following holds:*

$$\mathbb{E}\left[\sup_{\mathbb{X} \in \mathcal{H}_{\mathcal{X}}} |NL(\mathbb{X}|\mathcal{G}) - \mathbb{E}[NL(\mathbb{X}|.)]|\right] \leq \frac{4}{|\mathcal{O}|} + M\sqrt{\frac{|\mathcal{G}^*|}{|\mathcal{R}||\mathcal{O}|^2}} \cdot \sqrt{\frac{d\ln\left(4eDK|\mathcal{O}|\sqrt{d}\right)}{|\mathcal{O}|}}.$$

*Proof.* Let $\mathcal{B}$ be a finite set of vectors of dimension $d$ such that if we place centers of balls of radius $\rho$ in all these points then these balls will cover the set $\mathcal{X}$. Using Lemma 11, we know that it is possible to find such a set $\mathcal{B}$ satisfying

$$|\mathcal{B}| \leq \left((2D\sqrt{d})/\rho\right)^d \tag{3}$$

because the diameter of the set $\mathcal{X}$ is at most $D$ (due to the assumptions of the theorem) and the dimension of the vectors in $\mathcal{X}$ is $d$. Next we assume that we are only searching for a maximum-likelihood solution $\mathbb{X}$ consisting of vectors from $\mathcal{B}$ so that we could use Lemma 10 which expects a finite hypothesis set $\mathcal{H}$. Even though we have so far been thinking of the vector representations $\mathbb{X}$ as lookup tables, it will be more convenient here to think of them as lists of vectors. Specifically, we assume that the objects and relations are ordered arbitrarily. The hypothesis set will then be represented as $\mathcal{H}_{\mathcal{B}} = \mathcal{B}^{|\mathcal{O}|+|\mathcal{R}|}$. Hence, using (3), we obtain $|\mathcal{H}_{\mathcal{B}}| \leq \left((2D\sqrt{d})/\rho\right)^{d(|\mathcal{O}|+|\mathcal{R}|)}$. Then from Lemma 10 we have:

$$\mathbb{E}\left[\sup_{\mathbb{X} \in \mathcal{B}} |L(\mathbb{X}|\mathcal{G}) - \mathbb{E}[L(\mathbb{X}|.)]|\right] \leq M\sqrt{\frac{|\mathcal{G}^*|\ln\left(2e|\mathcal{H}_{\mathcal{B}}|\right)}{2}}. \tag{4}$$

Using the bound on $|\mathcal{H}_{\mathcal{B}}|$, we can bound the r.h.s. by:

$$M\sqrt{\frac{|\mathcal{G}^*|d(|\mathcal{O}|+|\mathcal{R}|)\ln\left(\frac{4eD\sqrt{d}}{\rho}\right)}{2}} \tag{5}$$

For fixed vector representations $\mathbb{X}'$ of the objects and relations, let $\mathcal{H}(\mathbb{X}')$ denote the set of vector representations $\mathbb{X}$ which satisfy that for every object or relation $o$: $\|\mathbf{x}_o - \mathbf{x}'_o\| \leq \rho$ where $\mathbf{x}_o$ and $\mathbf{x}'_o$ are vector representations of $o$ from $\mathbb{X}$ and $\mathbb{X}'$, respectively. Now, optimizing over the hypothesis set $\mathcal{H}_{\mathcal{X}}$ of all possible vector representations from $\mathcal{X}$ instead of just the vectors from the set $\mathcal{H}_{\mathcal{B}}$, would

yield the next bound:

$$\mathbb{E}\left[\sup_{\mathbb{X}\in\mathcal{H}_{\mathcal{X}}}|L(\mathbb{X}|\mathcal{G})-\mathbb{E}[L(\mathbb{X}|.)]|\right]$$

$$=\mathbb{E}\left[\sup_{\mathbb{X}'\in\mathcal{H}_{\mathcal{B}}}\sup_{\mathbb{X}\in\mathcal{H}(\mathbb{X}')}|L(\mathbb{X}|\mathcal{G})-\mathbb{E}[L(\mathbb{X}'|.)]+\mathbb{E}[L(\mathbb{X}'|.)]-\mathbb{E}[L(\mathbb{X}|.)]|\right]$$

$$\leq\mathbb{E}\left[\sup_{\mathbb{X}'\in\mathcal{H}_{\mathcal{B}}}\sup_{\mathbb{X}\in\mathcal{H}(\mathbb{X}')}|L(\mathbb{X}|\mathcal{G})-L(\mathbb{X}'|\mathcal{G})+L(\mathbb{X}'|\mathcal{G})-\mathbb{E}[L(\mathbb{X}'|.)]|\right]$$
$$+\sup_{\mathbb{X}'\in\mathcal{H}_{\mathcal{B}}}\sup_{\mathbb{X}\in\mathcal{H}(\mathbb{X}')}|\mathbb{E}[L(\mathbb{X}'|.)]-\mathbb{E}[L(\mathbb{X}|.)]|$$

$$\leq\mathbb{E}\left[\sup_{\mathbb{X}'\in\mathcal{H}_{\mathcal{B}}}\sup_{\mathbb{X}\in\mathcal{H}(\mathbb{X}')}|L(\mathbb{X}|\mathcal{G})-L(\mathbb{X}'|\mathcal{G})|\right]+\mathbb{E}\left[\sup_{\mathbb{X}'\in\mathcal{H}_{\mathcal{B}}}|L(\mathbb{X}'|\mathcal{G})-\mathbb{E}[L(\mathbb{X}'|.)]|\right]$$
$$+\sup_{\mathbb{X}'\in\mathcal{H}_{\mathcal{B}}}\sup_{\mathbb{X}\in\mathcal{H}(\mathbb{X}')}|\mathbb{E}[L(\mathbb{X}'|.)]-\mathbb{E}[L(\mathbb{X}|.)]|$$

Next, to bound the first and third term, we use Lemma 9 and, to bound the second term, we use Equations (4) and (5). After setting $\rho:=\frac{1}{K|\mathcal{O}|}$ and after simplifying, this gives us the following bound:

$$\mathbb{E}\left[\sup_{\mathbb{X}\in\mathcal{H}_{\mathcal{X}}}|L(\mathbb{X}|\mathcal{G})-\mathbb{E}[L(\mathbb{X}|.)]|\right]\leq 4|\mathcal{O}||\mathcal{R}|+M\sqrt{\frac{|\mathcal{G}^*|d(|\mathcal{O}|+|\mathcal{R}|)\ln\left(4eDK|\mathcal{O}|\sqrt{d}\right)}{2}}$$
$$\leq 4|\mathcal{O}||\mathcal{R}|+M\sqrt{|\mathcal{G}^*||\mathcal{O}||\mathcal{R}|d\ln\left(4eDK|\mathcal{O}|\sqrt{d}\right)}.$$

For the expected error of normalized log-likelihood we then obtain

$$\mathbb{E}\left[\sup_{\mathbb{X}\in\mathcal{H}_{\mathcal{X}}}|NL(\mathbb{X}|\mathcal{G})-\mathbb{E}[NL(\mathbb{X}|.)]|\right]\leq\frac{4}{|\mathcal{O}|}+M\sqrt{\frac{|\mathcal{G}^*|}{|\mathcal{R}||\mathcal{O}|^2}}\cdot\sqrt{\frac{d\ln\left(4eDK|\mathcal{O}|\sqrt{d}\right)}{|\mathcal{O}|}}$$

which finishes the proof of the theorem.

$\square$

## D  PROOF OF THEOREM 3

In this section, we prove Theorem 3.

**Theorem 3.** *Let $\mathcal{G}^*$ be the ground-truth graph on a set of objects $\mathcal{O}$ and relations $\mathcal{R}$. Let $P$ denote the data-generating distribution induced by $\mathcal{G}^*$ with the missingness parameter $\delta$ and let $Q$ denote the learned distribution. Let $t\in(0;1-\delta)$ be a threshold for predicting which triples belong to the knowledge graph. Finally, let $\mathcal{F}$ be the set of incorrectly predicted triples, i.e. the set of all triples $\tau\in\mathcal{O}\times\mathcal{R}\times\mathcal{O}$ such that either $\tau\in\mathcal{G}^*$ and the probability of $\tau$ given by the learned distribution $Q$ is smaller than $t$ (false negatives) or $\tau\notin\mathcal{G}^*$ and the probability of $\tau$ given by the learned distribution $Q$ is greater or equal to $t$ (false positives). Then the following holds for the cardinality of the set of incorrectly predicted triples $\mathcal{F}$:*

$$|\mathcal{F}|\leq\max\left\{\frac{\ln 2}{2t^2},\frac{\ln 2}{2(1-\delta-t)^2}\right\}\cdot KL(P||Q).$$

In the proof of the theorem, we will use the following simple lemma (which can be verified by computing the discriminant of the respective quadratic equation).

**Lemma 12.** *The following inequality holds for all $x,y\in[0;1]^2$ and $\alpha\in(0;\infty)$:*

$$\alpha(x-y)^2+\frac{1}{4\alpha}\geq|x-y|.$$

*Proof of Theorem 3.* We assume that all possible triples $(h, r, t) \in \mathcal{O} \times \mathcal{R} \times \mathcal{O}$ are ordered (arbitrarily) and that $\tau_i$ denotes the $i$-th triple from this set and $\mathcal{I} = \{1, 2, \ldots, |\mathcal{O}|^2 \cdot |\mathcal{R}|\}$. Let $\mathcal{G}$ be sampled from the data-generating distribution and $\mathcal{G}'$ be sampled from the learned model (i.e. the KG-distribution). We define binary random variables $B_i := \mathbb{1}(\tau_i \in \mathcal{G})$ and $B_i' := \mathbb{1}(\tau_i \in \mathcal{G}')$. Hence the complete vectors $\mathbf{B} = (B_1, \ldots, B_n)$ and $\mathbf{B}' = (B_1', \ldots, B_n')$ represent the knowledge graphs sampled from the data-generating and modelled distributions, respectively. We denote by $P_{\mathbf{B}}$ and $P_{\mathbf{B}}'$ the distributions of $\mathbf{B}$ and $\mathbf{B}'$. Due to the independence properties of the two distributions, the following holds: $KL(P_{\mathbf{B}}||P_{\mathbf{B}'}) = \sum_{i \in \mathcal{I}} KL(P_{B_i}||P_{B_i'})$ where $P_{B_i}$ and $P_{B_i}'$ are the distributions of $B_i$ and $B_i'$, respectively.

Next we bound the error that is made when we classify all triples $\tau_i$ with $P[B_i' = 1] > t$ as true (i.e. belonging to the knowledge graph) and the rest as false. We may notice that when misclassifying a triple $\tau_i$, we must either have

$$P[B_i = 1] = 0 \text{ and } P[B_i' = 1] > t$$

("false positive") or

$$P[B_i = 1] = 1 - \delta \text{ and } P[B_i' = 1] < t$$

("false negative"). Hence, if we denote by $\mathcal{FP}$ the set of triples $\tau_i$ for which we have false positive errors, we must have for all $\alpha > 0$

$$t|\mathcal{FP}| \leq \sum_{\tau_i \in \mathcal{FP}} P[B_i' = 1] \leq \sum_{\tau_i \in \mathcal{FP}} (|P[B_i = 1] - P[B_i' = 1]|)$$

$$\leq \sum_{\tau_i \in \mathcal{FP}} \left( \frac{1}{4\alpha} + \alpha |P[B_i = 1] - P[B_i' = 1]|^2 \right)$$

where the last inequality follows from Lemma 12. It follows that

$$|\mathcal{FP}| \cdot \left( t - \frac{1}{4\alpha} \right) \leq \sum_{\tau_i \in \mathcal{FP}} \alpha |P[B_i = 1] - P[B_i' = 1]|^2.$$

Letting $\alpha = 1/(2t)$ yields the bound

$$|\mathcal{FP}| \leq \frac{1}{t^2} \cdot \sum_{\tau_i \in \mathcal{FP}} |P[B_i = 1] - P[B_i' = 1]|^2. \tag{6}$$

Similarly for false negatives, if we denote by $\mathcal{FN}$ the set of triples for which we have false negative errors, we must have for all $\alpha' > 0$:

$$|\mathcal{FN}| \cdot (1 - \delta - t) \leq \sum_{\tau_i \in \mathcal{FN}} |P[B_i = 1] - P[B_i' = 1]|$$

$$\leq \sum_{\tau_i \in \mathcal{FN}} \left( \frac{1}{4\alpha'} + \alpha' |P[B_i = 1] - P[B_i' = 1]|^2 \right).$$

Hence, we also have:

$$|\mathcal{FN}| \cdot \left( \frac{4\alpha' - 1}{4\alpha'} - \delta - t \right) \leq \alpha' \sum_{\tau_i \in \mathcal{FN}} |P[B_i = 1] - P[B_i' = 1]|^2$$

Letting $\alpha' = 1/(2 - 2\delta - 2t)$ gives us

$$|\mathcal{FN}| \leq \frac{1}{(1 - \delta - t)^2} \cdot \sum_{\tau_i \in \mathcal{FN}} |P[B_i = 1] - P[B_i' = 1]|^2. \tag{7}$$

Next we combine (6) and (7) and apply Pinsker's inequality:

$$
\begin{aligned}
|\mathcal{FP}| + |\mathcal{FN}| &\leq \frac{1}{t^2} \cdot \sum_{\tau_i \in \mathcal{FP}} |P[B_i = 1] - P[B_i' = 1]|^2 \\
&\quad + \frac{1}{(1-\delta-t)^2} \cdot \sum_{\tau_i \in \mathcal{FN}} |P[B_i = 1] - P[B_i' = 1]|^2 \\
&= \frac{1}{t^2} \cdot \sum_{\tau_i \in \mathcal{FP}} \frac{1}{4} \|P_{B_i} - P_{B_i'}\|_1^2 + \frac{1}{(1-\delta-t)^2} \cdot \sum_{\tau_i \in \mathcal{FN}} \frac{1}{4} \|P_{B_i} - P_{B_i'}\|_1^2 \\
&\leq \frac{1}{4} \max \left\{ \frac{1}{t^2}, \frac{1}{(1-\delta-t)^2} \right\} \cdot \sum_{i \in \mathcal{I}} \|P_{B_i} - P_{B_i'}\|_1^2 \\
&\leq \max \left\{ \frac{\ln 2}{2t^2}, \frac{\ln 2}{2(1-\delta-t)^2} \right\} \cdot KL(P_{\mathbf{B}} \| P_{\mathbf{B}'}). \quad (8)
\end{aligned}
$$

This finishes the proof of this theorem.

$\square$

# E    PROOF OF THEOREM 4

**Theorem 4** *Let $\mathcal{G}^*$ be a ground-truth knowledge graph on a set of objects $\mathcal{O}$ and a set of relations $\mathcal{R}$ and let $\mathcal{G}$ be sampled by the data-generating distribution. Let $\mathcal{X} \subseteq \mathbb{R}^d$. Let $\mathcal{H}_{\mathcal{X}}$ ("hypothesis class") denote the set of vector representations $\mathbb{X}$, $\mathbb{X} \subseteq \mathcal{X}^{|\mathcal{O}|+|\mathcal{R}|}$, of the objects from $\mathcal{O}$ and relations from $\mathcal{R}$. Let $K$, $M$, $D$ be as in Theorem 2 and $t$ and $\delta$ be as in Theorem 3. Let $\mathbb{X} \in \mathcal{H}_{\mathcal{X}}$ be a representation of objects and relations learned by maximizing the log-likelihood $L(\mathbb{X}|\mathcal{G})$, i.e. $\mathbb{X} = \arg\max_{\mathbb{X}' \in \mathcal{H}_{\mathcal{X}}} L(\mathbb{X}'|\mathcal{G})$. Finally, let $\mathbb{X}^* = \arg\max_{\mathbb{X} \in \mathcal{H}_{\mathcal{X}}} \mathbb{E}[L(\mathbb{X}|.)]$. Then the following holds for the sets of incorrectly predicted triples $\mathcal{F}_{\mathbb{X}}$, obtained using the KG-distribution with $\mathbb{X}$:*

$$
\mathbb{E}\left[\frac{|\mathcal{F}_{\mathbb{X}}|}{|\mathcal{O}|^2 |\mathcal{R}|}\right] \leq \max\left\{\frac{\ln 2}{t^2}, \frac{\ln 2}{(1-\delta-t)^2}\right\} \cdot \left(\frac{4}{|\mathcal{O}|} \right.
$$
$$
\left. + M\sqrt{\frac{|\mathcal{G}^*|}{|\mathcal{R}||\mathcal{O}|^2}} \cdot \sqrt{\frac{d\ln\left(4eDK|\mathcal{O}|\sqrt{d}\right)}{|\mathcal{O}|} + \frac{KL(P\|P_{\mathbb{X}^*})}{|\mathcal{O}|^2 |\mathcal{R}|}} \right)
$$

*where $KL(P\|P_{\mathbb{X}^*})$ is the Kullback-Leibler divergence of the data-generating distribution $P$ and the best possible representable distribution $P_{\mathbb{X}^*}$.*

*Proof.* Let $\mathcal{G}$ denote the knowledge graph sampled from the data-generating distribution and $P_{\mathbb{X}}$ the learned distribution. For a fixed $\mathbb{X}_{fixed}$, the following equality holds: $KL(P\|P_{\mathbb{X}_{fixed}}) = H(P) + \mathbb{E}_P[L(\mathbb{X}_{fixed}|\mathcal{G})]$. Then we also have $KL(P\|P_{\mathbb{X}_{fixed}}) - KL(P\|P_{\mathbb{X}^*}) = \mathbb{E}_P[L(\mathbb{X}_{fixed}|\mathcal{G})] - \mathbb{E}_P[L(\mathbb{X}^*|\mathcal{G})]$. Next we can define $g(\mathcal{G}') := \arg\max_{\mathbb{X} \in \mathcal{H}_{\mathcal{X}}} L(\mathbb{X}|\mathcal{G}')$, substitute $g(\mathcal{G}')$ for $\mathbb{X}_{fixed}$ and take the expectation over graphs $\mathcal{G}'$, which are assumed to be sampled from the data-generating distribution. This yields:

$$
\mathbb{E}_{\mathcal{G}' \sim P}[KL(P\|P_{g(\mathcal{G}')}) - KL(P\|P_{\mathbb{X}^*})] = \mathbb{E}_{\mathcal{G}' \sim P}[\mathbb{E}_{\mathcal{G} \sim P}[L(g(\mathcal{G}')|\mathcal{G})] - \mathbb{E}_{\mathcal{G} \sim P}[L(\mathbb{X}^*|\mathcal{G})]].
$$

Next we can use Lemma 8.2 from (Devroye et al., 2013) to bound the r.h.s.:

$$
\mathbb{E}_{\mathcal{G}' \sim P}[\mathbb{E}_{\mathcal{G} \sim P}[L(g(\mathcal{G}')|\mathcal{G})] - \mathbb{E}_{\mathcal{G} \sim P}[L(\mathbb{X}^*|\mathcal{G})]] \leq 2\mathbb{E}_{\mathcal{G}' \sim P}\left[\sup_{\mathbb{X} \in \mathcal{H}_{\mathcal{X}}} |L(\mathbb{X}|\mathcal{G}') - \mathbb{E}[L(\mathbb{X}, .)]|\right].
$$

Finally, we use the bound from Theorem 2 to bound the r.h.s. of this inequality. Combining this with Theorem 3 finishes the proof.    $\square$

# F    PROOF OF THEOREM 5

**Theorem 5.** *Let $\mathcal{G}^*$ be the ground-truth graph on a set of objects $\mathcal{O}$ and relations $\mathcal{R}$. Let $P$ denote the data-generating distribution induced by $\mathcal{G}^*$ and let $P_\gamma$ be a $\gamma$-admissible distribution. Let $Q$*

*denote the learned distribution and let $t \in (\gamma; 1 - \gamma)$ be a threshold. Then the following holds for the cardinality of the set of incorrectly predicted triples $\mathcal{F}$:*

$$|\mathcal{F}| \leq \max \left\{ \frac{\ln 2}{2(t - \gamma)^2}, \frac{\ln 2}{2(1 - \gamma - t)^2} \right\} \cdot KL(P_\gamma || Q).$$

*Proof. The proof of this theorem is very similar to the proof of Theorem 3. We highlight the most important differences in bold.*

We assume that all possible triples $(h, r, t) \in \mathcal{O} \times \mathcal{R} \times \mathcal{O}$ are ordered (arbitrarily) and that $\tau_i$ denotes the $i$-th triple from this set and $\mathcal{I} = \{1, 2, \ldots, |\mathcal{O}|^2 \cdot |\mathcal{R}|\}$. **Let $\mathcal{G}$ be sampled from the distribution $\mathbf{P}_\gamma$** and $\mathcal{G}'$ be sampled from the learned model (i.e. the KG-distribution). We define binary random variables $B_i := \mathbb{1}(\tau_i \in \mathcal{G})$ and $B_i' := \mathbb{1}(\tau_i \in \mathcal{G}')$. Hence the complete vectors $\mathbf{B} = (B_1, \ldots, B_n)$ and $\mathbf{B}' = (B_1', \ldots, B_n')$ represent the knowledge graphs sampled from the distribution $P_\gamma$ and the modelled distribution, respectively. We denote by $P_\mathbf{B}$ and $P_\mathbf{B}'$ the distributions of $\mathbf{B}$ and $\mathbf{B}'$. Due to the independence properties of the two distributions (**after all, $\mathbf{P}_\gamma$ is still a KG-distribution and therefore inherits its independence structure**), the following holds: $KL(P_\mathbf{B} || P_\mathbf{B'}) = \sum_{i \in \mathcal{I}} KL(P_{B_i} || P_{B_i'})$ where $P_{B_i}$ and $P_{B_i}'$ are the distributions of $B_i$ and $B_i'$, respectively.

Next we bound the error that is made when we classify all triples $\tau_i$ with $P[B_i' = 1] > t$ as true (i.e. belonging to the knowledge graph) and the rest as false. We may notice that when misclassifying a triple $\tau_i$, we must either have

$$\mathbf{P[B_i = 1] \leq \gamma} \text{ and } P[B_i' = 1] > t$$

("false positive") or

$$\mathbf{P[B_i = 1] \geq 1 - \gamma} \text{ and } P[B_i' = 1] < t$$

("false negative").

*The rest of differences w.r.t. the proof of Theorem 3 follows from the differences highlighted so far in a straightforward way (details follow below).*

Hence, if we denote by $\mathcal{FP}$ the set of triples $\tau_i$ for which we have false positive errors, we must have for all $\alpha > 0$

$$t|\mathcal{FP}| \leq \sum_{\tau_i \in \mathcal{FP}} P[B_i' = 1] \leq \sum_{\tau_i \in \mathcal{FP}} (\gamma + |P[B_i = 1] - P[B_i' = 1]|)$$

$$\leq \sum_{\tau_i \in \mathcal{FP}} \left( \gamma + \frac{1}{4\alpha} + \alpha |P[B_i = 1] - P[B_i' = 1]|^2 \right)$$

where the last inequality follows from Lemma 12. It follows that

$$|\mathcal{FP}| \cdot \left( t - \gamma - \frac{1}{4\alpha} \right) \leq \sum_{\tau_i \in \mathcal{FP}} \alpha |P[B_i = 1] - P[B_i' = 1]|^2.$$

Letting $1/(2t - 2\gamma)$ yields the bound

$$|\mathcal{FP}| \leq \frac{1}{(t - \gamma)^2} \cdot \sum_{\tau_i \in \mathcal{FP}} |P[B_i = 1] - P[B_i' = 1]|^2. \tag{9}$$

Similarly for false negatives, if we denote by $\mathcal{FN}$ the set of triples for which we have false negative errors, we must have for all $\alpha' > 0$:

$$|\mathcal{FN}| \cdot (1 - \gamma - t) \leq \sum_{\tau_i \in \mathcal{FN}} |P[B_i = 1] - P[B_i' = 1]|$$

$$\leq \sum_{\tau_i \in \mathcal{FN}} \left( \frac{1}{4\alpha'} + \alpha' |P[B_i = 1] - P[B_i' = 1]|^2 \right).$$

Hence, we also have:

$$|\mathcal{FN}| \cdot \left( \frac{4\alpha' - 1}{4\alpha'} - \gamma - t \right) \leq \alpha' \sum_{\tau_i \in \mathcal{FN}} |P[B_i = 1] - P[B_i' = 1]|^2$$

Letting $\alpha' = 1/(2 - 2\gamma - 2t)$ gives us

$$|\mathcal{FN}| \leq \frac{1}{(1 - \gamma - t)^2} \cdot \sum_{\tau_i \in \mathcal{FN}} |P[B_i = 1] - P[B_i' = 1]|^2. \tag{10}$$

Next, we combine (9) and (10) and apply Pinsker's inequality:

$$
\begin{aligned}
|\mathcal{FP}| + |\mathcal{FN}| &\leq \frac{1}{(t - \gamma)^2} \cdot \sum_{\tau_i \in \mathcal{FP}} |P[B_i = 1] - P[B_i' = 1]|^2 \\
&\quad + \frac{1}{(1 - \gamma - t)^2} \cdot \sum_{\tau_i \in \mathcal{FN}} |P[B_i = 1] - P[B_i' = 1]|^2 \\
&= \frac{1}{(t - \gamma)^2} \cdot \sum_{\tau_i \in \mathcal{FP}} \frac{1}{4} \|P_{B_i} - P_{B_i'}\|_1^2 + \frac{1}{(1 - \gamma - t)^2} \cdot \sum_{\tau_i \in \mathcal{FN}} \frac{1}{4} \|P_{B_i} - P_{B_i'}\|_1^2 \\
&\leq \frac{1}{4} \max \left\{ \frac{1}{(t - \gamma)^2}, \frac{1}{(1 - \gamma - t)^2} \right\} \cdot \sum_{i \in \mathcal{I}} \|P_{B_i} - P_{B_i'}\|_1^2 \\
&\leq \max \left\{ \frac{\ln 2}{2(t - \gamma)^2}, \frac{\ln 2}{2(1 - \gamma - t)^2} \right\} \cdot KL(P_{\mathbf{B}} \| P_{\mathbf{B}'}). \quad (11)
\end{aligned}
$$

This finishes the proof of this theorem. $\qquad\square$

## G   PROOF OF THEOREM 6

**Theorem 6** *Let everything be as in Theorem 4 but assume a $\gamma$-realizable setting. Then the following holds for the sets of incorrectly predicted triples $\mathcal{F}_{\mathbb{X}}$, obtained using the KG-distribution with $\mathbb{X}$:*

$$\mathbb{E} \left[ \frac{|\mathcal{F}_{\mathbb{X}}|}{|\mathcal{O}|^2 |\mathcal{R}|} \right] \leq \max \left\{ \frac{\ln 2}{(t - \gamma)^2}, \frac{\ln 2}{(1 - t - \gamma)^2} \right\} \cdot \left( \frac{4}{|\mathcal{O}|} + M \sqrt{\frac{|\mathcal{G}^*|}{|\mathcal{R}||\mathcal{O}|^2}} \sqrt{\frac{d \ln \left( 4eDK|\mathcal{O}|\sqrt{d} \right)}{|\mathcal{O}|}} \right)$$

*Proof.* The proof of this theorem is practically identical to the proof of Theorem 4. The only difference is that in this proof we invoke Theorem 5 instead of Theorem 3. $\qquad\square$

## H   THE LIPSCHITZ CONSTANT OF SIMPLE'S SCORING FUNCTION

In this section we derive a bound on the Lipschitz constant $K$ of the scoring function of SimplE (Kazemi & Poole, 2018) on a bounded domain $\mathcal{X} \subseteq \mathbb{R}^{2d}$.[3] The scoring function of SimplE is $\psi(\mathbf{x}_o, \mathbf{x}_r, \mathbf{x}_t) = \frac{1}{2} \left( \langle \mathbf{x}_h', \mathbf{x}_r', \mathbf{x}_t'' \rangle + \langle \mathbf{x}_t', \mathbf{x}_r'', \mathbf{x}_h'' \rangle \right)$ where $\langle \mathbf{x}, \mathbf{y}, \mathbf{z} \rangle := \sum_{i=1}^{d} x_i y_i z_i$. Let us denote $\mathbf{x}_h = [\mathbf{x}_h', \mathbf{x}_h'']$, $\mathbf{x}_r = [\mathbf{x}_r', \mathbf{x}_r'']$, $\mathbf{x}_t = [\mathbf{x}_t', \mathbf{x}_t'']$ and $\mathbf{r} = [\mathbf{x}_h, \mathbf{x}_r, \mathbf{x}_t]$. Since the function $\psi(\mathbf{r})$ is differentiable everywhere, any $K$ satisfying $K \geq \|\nabla \psi(\mathbf{r})\|$ for all $\mathbf{r} \in \mathcal{X}^3$ will be a Lipschitz constant for $\psi$ on the bounded domain $\mathcal{X}$.

Next we compute the gradient of $\psi(\mathbf{r})$. To simplify the notation, we use $\mathbf{x}[i]$ to denote the $i$-th element of the vector $\mathbf{x}$. We then have

---

[3] Here we use $d$ to denote the dimension of the vector representations as used by Kazemi & Poole (2018). Since SimplE defines two vectors for every object or relation, the dimension in our formalism is $2d$, hence $\mathcal{X} \subseteq \mathbb{R}^{2d}$.

$$\frac{\partial \psi(\mathbf{r})}{\partial \mathbf{x}_h'[i]} = \frac{1}{2} \cdot \mathbf{x}_r'[i] \cdot \mathbf{x}_t''[i], \qquad \frac{\partial \psi(\mathbf{r})}{\partial \mathbf{x}_r'[i]} = \frac{1}{2} \cdot \mathbf{x}_h'[i] \cdot \mathbf{x}_t''[i], \qquad \frac{\partial \psi(\mathbf{r})}{\partial \mathbf{x}_t''[i]} = \frac{1}{2} \cdot \mathbf{x}_h'[i] \cdot \mathbf{x}_r'[i],$$

$$\frac{\partial \psi(\mathbf{r})}{\partial \mathbf{x}_t'[i]} = \frac{1}{2} \cdot \mathbf{x}_r''[i] \cdot \mathbf{x}_h''[i], \qquad \frac{\partial \psi(\mathbf{r})}{\partial \mathbf{x}_r''[i]} = \frac{1}{2} \cdot \mathbf{x}_t'[i] \cdot \mathbf{x}_h''[i], \qquad \frac{\partial \psi(\mathbf{r})}{\partial \mathbf{x}_h''[i]} = \frac{1}{2} \cdot \mathbf{x}_t'[i] \cdot \mathbf{x}_r''[i].$$

From this we get

$$\|\nabla \psi(\mathbf{r})\|^2 = \frac{1}{4} \sum_{i=1}^{d} \left( \mathbf{x}_r'[i]^2 \cdot \mathbf{x}_t''[i]^2 + \mathbf{x}_h'[i]^2 \cdot \mathbf{x}_t''[i]^2 + \mathbf{x}_h'[i]^2 \cdot \mathbf{x}_r'[i]^2 + \mathbf{x}_r''[i]^2 \cdot \mathbf{x}_h''[i]^2 \right.$$

$$\left. + \mathbf{x}_t'[i]^2 \cdot \mathbf{x}_h''[i]^2 + \mathbf{x}_t'[i]^2 \cdot \mathbf{x}_r''[i]^2 \right)$$

$$\leq \frac{1}{4} \left( \langle \mathbf{x}_r', \mathbf{x}_t'' \rangle^2 + \langle \mathbf{x}_h', \mathbf{x}_t'' \rangle^2 + \langle \mathbf{x}_h', \mathbf{x}_r' \rangle^2 + \langle \mathbf{x}_r'', \mathbf{x}_h'' \rangle^2 + \langle \mathbf{x}_t', \mathbf{x}_h'' \rangle^2 + \langle \mathbf{x}_t', \mathbf{x}_r'' \rangle^2 \right) \leq \frac{6}{4} \sup_{\mathbf{x} \in \mathcal{X}} \|\mathbf{x}\|^4.$$

Thus, we get

$$\|\nabla \psi(\mathbf{r})\| \leq \frac{\sqrt{6}}{2} \sup_{\mathbf{x} \in \mathcal{X}} \|\mathbf{x}\|^2 := K,$$

Let us now denote $\mathbf{y}_h = [\mathbf{y}_h', \mathbf{y}_h''], \mathbf{y}_r = [\mathbf{y}_r', \mathbf{y}_r''], \mathbf{y}_t = [\mathbf{y}_t', \mathbf{y}_t'']$, and $\mathbf{s} = [\mathbf{y}_h, \mathbf{y}_r, \mathbf{y}_t]$. Then we have

$$\|\psi(\mathbf{r}) - \psi(\mathbf{s})\| \leq K \|\mathbf{r} - \mathbf{s}\| \leq K \cdot \left( \|\mathbf{x}_h - \mathbf{y}_h\| + \|\mathbf{x}_r - \mathbf{y}_r\| + \|\mathbf{x}_t - \mathbf{y}_t\| \right).$$

So $K = \frac{\sqrt{6}}{2} \sup_{\mathbf{x} \in \mathcal{X}} \|\mathbf{x}\|^2$ is indeed the Lipschitz constant that we need in order to be able to obtain generalization bounds using Theorems 4 and 5.

