# OpenReview forum: "Knowledge Graph Embedding: A Probabilistic Perspective and Generalization Bounds"
_ICLR.cc/2020/Conference — Reject_

### Official Review · AnonReviewer1 · 2019-10-21
**Official Blind Review #1**

**Rating:** 6

**Review:**

The paper presents a study on bound of the number of the expected triples wrongly predicted by general graph embedding methods. The paper considers methods that follow the "completely at random" assumption and aim to maximize the likelihood (or minimize the log-loss).

The study considers the learning from positive and unlabelled data problem and theoretically demonstrates the correctness of the bounds on the number of triples that embedding methods can add during the completion of the knowledge graph.

The paper is well written, theorems and proofs seem to me to be mathematically sound and correct. To the best of my knowledge, I agree with the authors when they claim to be the first to inspect the problem to give bounds under the setting they have considered. However, I have to admit that I am not an expert on the problem, I am more aware of the practical part of this field and less aware of the theoretical part.

However, I have carefully checked the validity of the analysis and I have not found any flawless or critical errors in the proofs. I would suggest removing aliases for |O|, |R|, etc. from the proofs as they do not simplify the formulas significantly.

Other corrections:
- on page 5 the word "the" is written twice in the sentence just before Theorem 2.
- on page 7 C is defined as constant but C should be C_1. Also C_2 should be defined.
- on page 7, in "Lipschitz Continuity" after the sentence "Our result requires Lipschitz continuity" there must be a full stop or the word Most must begin with a lowercase m.
- on page 8 there is inside the Figure 1 (on the red line, near 1*10^4 number of objects in the domain |O|) a link to page 6 that should be removed.

**Experience Assessment:**

I do not know much about this area.

**Review Assessment: Checking Correctness Of Derivations And Theory:**

I assessed the sensibility of the derivations and theory.

**Review Assessment: Checking Correctness Of Experiments:**

N/A

**Review Assessment: Thoroughness In Paper Reading:**

N/A

---

> ### Author Response · Authors · 2019-11-13
> **Author response**
>
> Thank you for the review!
>
>
> > I would suggest removing aliases for |O|, |R|, etc.
> > from the proofs as they do not simplify the formulas significantly.
> > Other corrections:
> > - on page 5 the word "the" is written twice in the sentence just before Theorem 2.
> > - on page 7 C is defined as constant but C should be C_1. Also C_2 should be defined.
> ->  on page 7, in "Lipschitz Continuity" after the sentence "Our result requires Lipschitz
> > continuity" there must be a full stop or the word Most must begin with a lowercase m.
> > - on page 8 there is inside the Figure 1 (on the red line, near 1*10^4 number of objects in
> > the domain |O|) a link to page 6 that should be removed.
>
> We have fixed all of these in the revision. Thank you!

---

### Official Review · AnonReviewer3 · 2019-10-23
**Official Blind Review #3**

**Rating:** 1

**Review:**

Summary

This paper studied the knowledge completion problem from the viewpoint of statistical learning theory. More specifically, it formalized a knowledge graph completion problem as an estimation of the optimal embedding by assuming the canonical distribution derived from the embedding and derived the upper bound of misclassification rate of entities under the missing-completely-at-random assumption.


Decision

Although the formalization of knowledge graph completion problems is novel (to the best of my knowledge), there is much room for improvement in the organization of the paper. Therefore, I would recommend to reject the paper this time and to ask authors to revise the paper so that the paper is more accessible to researchers and engineers.
There are numerous existing works about the knowledge graph embedding problem and the knowledge graph completion problem using embeddings. However, few works have been done to justify these methods theoretically. This paper attempted to answer this question via statistical learning theorem perspectives. The theorems gave sufficient conditions in terms of the size of a knowledge graph and properties of embedding functions under which the misclassification rate goes to $0$. In this aspect, this paper gave insights into how we can give justifications to existing embedding methods.
However,  I think it is hard for those who are not familiar with this field to follow the logic of the paper, as I write in detail in the following sections. I understand some people argue that the paper organization is not essential for how a paper contributes to science and technology. However, the accessibility to the paper is vital to promote technical communications in different fields. Also, I believe it is beneficial for the paper to maximize its value. Thus, I would ask the authors to polish the paper.


Suggestions

- Introduction
	- Please explain what knowledge graphs are in the introduction before explaining how knowledge graphs represent knowledge, or what problems of practically available knowledge graphs have.
	- I think it is a bit too casual and conversational to use wordings like "Please bare in mind" in papers.
	- I think it is helpful if the authors add a summary of theorems and its implication in the introduction to grasp the theoretical contributions the paper has made.
- Section 6, Theorem 1
	- Please describe what is the probability of $P$ and with respect to which probability distribution $\mathbb{E}$ takes the expectation.
- Section 7
	- The authors claim that Pinsker's inequality plays a central role in deriving the upper bound for the ratio of wrongly predicted triples. However, we cannot know how the authors used the inequality unless we see the proof of Theorem 3, which is available in the appendix.
	- In Theorem 3, the definition of "wrongly predicted triple" is missing. I only find a description related to it in the proof of Theorem 3 in the appendix. Could you add the definition in the main part?
- Section 8, Theorem 4
	- The statement of Theorem 4 says that $\mathbb{X}$ is a learned representation. But it is not available in the main article how we obtain it. We know that the authors used the maximum-likelihood estimator if we read the proof of the theorem. The author should clarify it in the statement of the theorem.
- Section 10
	- What does "this" mean in the title of the section?
	- In the initial reading, I could not understand the main point the authors want to address in the paragraph starting with Loss. I think authors can clarify the point by restructuring the paragraph.


Minor comments

- Section 2 and Section 6
	- The authors treated $\mathbb{X}$, which is the set of vector representations of objects and relations, as a lookup table in Section 2.2 and reinterpreted it as the subset of $\mathcal{X}^{|\mathcal{O}|+|\mathcal{R}|}$ in Section 6. I think we can simplify the description if we treat $\mathbb{X}$ as the subset of $\mathcal{X}^{\mathcal{O} \sqcup \mathcal{R}} := \{f: \mathcal{O} \sqcup \mathcal{R} \to \mathcal{X} \}$ and denote $\mathbb{X}(o)$ as $x_o$ in short hand for a representation $\mathbb{X}$ and an object $o\in \mathcal{O}$ (and similarly for $x_r$).
- Section 4
	- It would be helpful to make the correspondence explicit between each sentence and theorem later in the last paragraph. For example, "We prove a generalization bound for log-likelihood from which a bound on Kullback-Leibler divergence follows (Theorem 3)" or something like that. - Section 6, Theorem 1
	- The definitions of $\mathbb{X}_o$ and $\mathbb{X}_r$ are missing, if I do not miss something.
	- It is better to add the assumption that $\sup_{h, r, t} |\psi(x_h, x_r, x_t)|$ is finite.
- Section 12
	- "simplistic" means too simple by itself. So, "too simplistic" should be "too simple" or "simplistic".


Questions

- Section 4
	- I could not fully understand the intuition of the proof in the second paragraph. The authors think that the situation is desirable if multiple knowledge graphs are available. However, in the first approach, they concatenated the graphs into one (I interpreted the union of graphs as $\widehat{\mathcal{G}}_1 \cup \widehat{\mathcal{G}}_2 \cup \cdots \widehat{\mathcal{G}}_n$. Correct me if I am wrong). Since this operation would reduce the situation to the single-graph case, we cannot take advantage of multiple graphs.
- Section 7, Theorem 4
	- The upper bound is in terms of the estimator which maximizes the expected log-likelihood. I am wondering whether this estimator can achieve the best possible misclassification rate (i.e., $|\mathcal{F}/|\mathcal{O}^2|/|\mathcal{R}||$). I understand that it minimizes the discrepancy between the inferred distribution and the data generating distribution. But I am not sure it implies the least misclassification rate.
	- The small terms in the upper bound (i.e., first and second terms) depend on the number of objects $|\mathcal{O}|$ and not on relations $|\mathcal{R}|$. Since the role of $\mathcal{O}^2$ and $\mathcal{R}$ is symmetric mathematically (if I understand correctly), I feel it is weird that the additional terms do not necessarily go to zero when $\mathcal{R}\to \infty$. I particular, I expected that the additonal small terms depend solely on $|\mathcal{G}|=|\mathcal{O}|^2|\mathcal{R}|$.

**Experience Assessment:**

I have read many papers in this area.

**Review Assessment: Checking Correctness Of Derivations And Theory:**

I carefully checked the derivations and theory.

**Review Assessment: Checking Correctness Of Experiments:**

I carefully checked the experiments.

**Review Assessment: Thoroughness In Paper Reading:**

I read the paper at least twice and used my best judgement in assessing the paper.

---

> ### Author Response · Authors · 2019-11-13
> **Author response**
>
> Thank you for the very constructive feedback!
>
> We tried to incorporate your suggestions into the revision of our paper.
>
> > Suggestions
> >
> > - Introduction
> >	- Please explain what knowledge graphs are in the introduction before explaining
> > how knowledge graphs represent knowledge, or what problems of practically available
> > knowledge graphs have.
>
> We have rewritten the beginning of the introduction.
>
> > - I think it is a bit too casual and conversational to use wordings like "Please bare in
> > mind" in papers.
>
> We have reworded this sentence.
>
> > - I think it is helpful if the authors add a summary of theorems and its implication in the
> > introduction to grasp the theoretical contributions the paper has made.
>
> We have added a paragraph “Main Technical Contributions” at the end of the introduction section.
>
> > - Section 6, Theorem 1
> > - Please describe what is the probability of  and with respect to which probability
> > distribution  takes the expectation.
>
> Done.
>
> > - Section 7
> >	- The authors claim that Pinsker's inequality plays a central role in deriving the
> > upper bound for the ratio of wrongly predicted triples. However, we cannot know how the
> > authors used the inequality unless we see the proof of Theorem 3, which is available in
> > the appendix.
>
> We have expanded this section. Now we also give a sketch of the proof idea which explains how Pinsker’s inequality is used to obtain the result in this section.
>
> >	- In Theorem 3, the definition of "wrongly predicted triple" is missing. I only find a
> > description related to it in the proof of Theorem 3 in the appendix. Could you add the
> > definition in the main part?
>
> We have now added it in the text at the beginning of this section and we have also recalled it again in the statement of the theorem.
>
> > - Section 8, Theorem 4
> >	- The statement of Theorem 4 says that  is a learned representation. But it is not
> > available in the main article how we obtain it. We know that the authors used the
> > maximum-likelihood estimator if we read the proof of the theorem. The author should
> > clarify it in the statement of the theorem.
>
> It had actually been said in words in the statement of the theorem that X is obtained by maximizing log-likelihood, but it was probably confusing. We have added text before the theorem that describes how the vector representations are learned and also added an argmax L(X|G) to the statement of the theorem.
>
>
> > - Section 10
> >	- What does "this" mean in the title of the section?
>
> We have changed the name of the section.
>
> >	- In the initial reading, I could not understand the main point the authors want to
> > address in the paragraph starting with Loss. I think authors can clarify the point by
> > restructuring the paragraph.
>
> We have expanded the section and show the correspondence between our setting and existing settings explicitly in detail.
>
>
> > Minor comments
> >
> > - Section 2 and Section 6
> > 	- The authors treated , which is the set of vector representations of objects and
> > relations, as a lookup table in Section 2.2 and reinterpreted it as the subset of  in Section
> > 6. I think we can simplify the description if we treat  as the subset of  and denote  as  in
> > short hand for a representation  and an object  (and similarly for ).
>
> We agree that this would be more elegant mathematically but it might be a bit less intuitive (but that may be subjective). However, we are open to still implement it in the paper.
>
> > - Section 4
> >	- It would be helpful to make the correspondence explicit between each sentence
> > and theorem later in the last paragraph. For example, "We prove a generalization bound
> > for log-likelihood from which a bound on Kullback-Leibler divergence follows (Theorem
> > 3)” or something like that.
>
> Done.
>  > - Section 6, Theorem 1
> >	- The definitions of  and  are missing, if I do not miss something.
> Sorry, that was a relict from a previous version of the draft of our paper. It is fixed now.
>
> >	- It is better to add the assumption that  is finite.
>
> Done.
>
> > - Section 12
> >	- "simplistic" means too simple by itself. So, "too simplistic" should be "too simple"
> > or “simplistic".
>
> Done
>
>
> The response to your question is posted separately (we ran into character limit).

---

> ### Author Response · Authors · 2019-11-13
> **Author response (part 2)**
>
> > Questions
> >
> > - Section 4
> >	- I could not fully understand the intuition of the proof in the second paragraph. The
> > authors think that the situation is desirable if multiple knowledge graphs are available.
> > However, in the first approach, they concatenated the graphs into one (I interpreted the
> > union of graphs as . Correct me if I am wrong). Since this operation would reduce the
> > situation to the single-graph case, we cannot take advantage of multiple graphs.
>
> This example might have been confusing so we removed it from the revision. What we wanted to convey was the following idea. If we had a large number of samples from the same ground truth knowledge graph, each triple that is contained in the ground-truth knowledge graph would appear with high probability in the union of all the samples. What we wanted to say with this is that it is enough to learn the data-generating distribution (which is a proxy to the ground-truth knowledge graph). But as we realize now, it might have been confusing, so we removed it.
>
> > - Section 7, Theorem 4
> >	- The upper bound is in terms of the estimator which maximizes the expected log-
> > likelihood. I am wondering whether this estimator can achieve the best possible
> > misclassification rate (i.e., ). I understand that it minimizes the discrepancy between the
> > inferred distribution and the data generating distribution. But I am not sure it implies the
> > least misclassification rate.
>
> Indeed, we believe you are right, but our point in this paper is to obtain a new perspective and generalization bounds for something that people actually do in the knowledge graph embedding literature (e.g. ComplEx, SimplE, ConvE). Since these methods also maximize  log-likelihood which corresponds to minimizing the discrepancy (KL-divergence) of the data-generating and the modelled distribution, we need to bound the error in terms of these (else our analysis would not apply). Besides, it is normal in machine learning to use different losses than the 0-1 loss for computational complexity reasons.
>
> >	- The small terms in the upper bound (i.e., first and second terms) depend on the
> > number of objects  and not on relations . Since the role of  and  is symmetric
> > mathematically (if I understand correctly), I feel it is weird that the additional terms do not
> > necessarily go to zero when . I particular, I expected that the additonal small terms
> > depend solely on .
>
> This is an interesting observation. The answer is going to be yes and no. First, indeed we made a (small) mistake and forgot a $|R|$ term under the square root, so our bound was not as tight as possible. So in fact, the sparsity term in the bound should be $|\mathcal{G}^*|/(|\mathcal{O}|^2*|\mathcal{R}|)$ (we apologize for this and fixed it in the paper). On the other hand, it is not true that $\mathcal{R}$ and $\mathcal{O}^2$ are mathematically symmetric. To describe both $R$ and $O^2$ one needs |\mathcal{R}| vectors and $\sqrt{|\mathcal{O}^2|}$, i.e. $|\mathcal{O}|$, vectors, respectively. So in this respect (counting the number of parameters), $\mathcal{R}$ would behave more similarly to $\mathcal{O}$ rather than $\mathcal{O}^2$. On the other hand, $\mathcal{R}$ and $\mathcal{O}^2$ behave the same when it comes to the number of examples (positive + unlabeled), as this is indeed $|\mathcal{R}|\cdot |\mathcal{O}|^2$.
>
> To give an intuitive example. Consider two extreme cases $|\mathcal{R}| = 1$ and $|\mathcal{O}| \rightarrow \infty$ vs $|\mathcal{R}| \rightarrow \infty$ and $|\mathcal{O}| = 2$. In the first case, we can generalize which our bounds prove. However, in the second case we cannot really guarantee generalization (we have $2 |\mathcal{R}|$ examples and $d \cdot |\mathcal{R}|$ parameters).

---

> > ### Comment · AnonReviewer3 · 2019-11-13
> > **Thank you for reflecting my comments.**
> >
> > I thank the authors for considering my review comments seriously and appreciate the authors' efforts for improving the paper. Also, I thank authors for answering my questions. I will check the revised version and will ask the authors again if necessary.

---

> > ### Comment · AnonReviewer3 · 2019-11-14
> > **Rereading Section 4 and further clarification**
> >
> > I reread Section 4 and noticed that there were some incorrect points in my understanding. When I read this section for the first time, I forgot that a knowledge graph is represented as a set of triples. Rather, I imagined the "usual" graph $G=(V, E)$. Therefore, I interpreted $\hat{G}_1 \cup \cdots \cup \hat{G}_n$ as the concatenation of the graphs $(V_1 \sqcup \cdots \sqcup V_n, E_1 \sqcup \cdots \sqcup E_n)$. Now I understand what the authors said the "naive approach" in the previous version of the paper (although it has been deleted in the revised version).
> >
> > However, another question has come up why the naive approach does not work. If we read the definition of the data-generating distribution (definition 1) literally, the sampled graph $\hat{G}$ is always the subset of the true graph $G^\ast$ and there are no fear of false positives. Should the data-generating distribution add an edge which is missing in $\mathcal{G}^\ast$ with probability $\delta$ to $\hat{\mathcal{G}}$? I want to clarify this point and want the authors correct me if I am wrong.
> >
> >
> > Minor comments
> > - Please add what the acronym of KG stands for when it appears for the first time.
> > - Add the definition of $\mathcal{R}$ to Section 4.

---

> > > ### Author Response · Authors · 2019-11-14
> > > **Re: Rereading Section 4 and further clarification**
> > >
> > > Thank you for getting back to us so quickly!
> > >
> > > > However, another question has come up why the naive approach does not work. If we read the
> > > > definition of the data-generating distribution (definition 1) literally, the sampled graph  is always
> > > > the subset of the true graph  and there are no fear of false positives. I want to clarify this point
> > > > and want the authors correct me if I am wrong.
> > >
> > > That is correct. The footnote 2 in the original version of the paper referred to the more general case in which noisy triples could be added by the data-generating process as well and the task would be to still recover the original KG. It does not mean that we assume a perfect knowledge graph as input, only that we consider the task usually considered in the literature and that is knowledge graph "completion" (that is also what the evaluation protocols in the respective practical papers measure). It would certainly be interesting to consider the problem of knowledge graph "filtration" in which the task would be to also remove triples that do not belong to the knowledge graph, but that is something that would need some input from the practical/experimental side as well.
> > >
> > >
> > > > Minor comments
> > >
> > > We will update these minor issues in the revision shortly before the deadline (in case there is more feedback we want to update the revision once).

---

> > > ### Author Response · Authors · 2019-11-15
> > > **Re: Author's response 2**
> > >
> > > Sorry. We have just noticed that the reviewer's comment is slightly different now than what we received by email (we suspect there was an edit for which we did not get a notification from openreview). This concerns the sentence "Should the data-generating distribution add an edge which is missing in  with probability $\delta$ to $\widehat{\mathcal{G}}$?" which we did not see originally in the email.
> > >
> > > That would be a different setting in which one would also perform "filtration" of the knowledge graph rather than just "completion" (we were also hinting at this in the previous response but now we wanted to be more explicit seeing the sentence that we originally had not seen). One could modify our analysis to accomodate for such noise as well (we explain how below) but that would again be something slightly different than what is really measured when people evaluate KG embedding methods in the literature on the KG "completion" task (because there is usually no other way to evaluate which triples included in the training KG are wrong than by laborious human anotation).
> > >
> > > However, please notice that you could first have a ground-truth-0 graph $\mathcal{G}_0$ then add the noise (triples that are not in the original $\mathcal{G}_0$) and then call the new graph $\mathcal{G}^*$ "ground truth KG". You can then still apply our analysis to the ground truth KG with triples missing completely at random and everything will work correctly as in our paper. We will just not be recovering $\mathcal{G}_0$ but $\mathcal{G}^*$ - after all KG completion is usually about adding triples.
> > >
> > > Now, how one could extend our results to KG-completion+filtration. Most of our analysis would carry on unchanged. One critical step is in the application of McDiarmid inequality. If we assumed that every triple can be added with some small probability, we would not be able to get the sparsity term in the resulting bounds for the expected number of errors. This problem can be solved by a simple modification of the noise model: sample M triples with replacement ($M \approx \delta' |\mathcal{O}|^2 |\mathcal{R}|$) from the complement of the knowledge graph and add them to $\mathcal{G}$ (instead of adding each possible triple independently with probability $\delta'$). For practical purposes these two settings are very similar, but this one is a bit easier to analyze. Another place where our analysis would need to be changed is the derivation of the bound depending on the KL-divergence. There we would have to replace at some places $P[B_i = 1] = 0$ by $P[B_i = 1] = \delta'$, but the rest of the analysis should not change much.
> > >
> > > We were trying to keep as close to the settings reported in the practical papers as possible which is why we did not consider KG "filtration".

---

> > > > ### Comment · AnonReviewer3 · 2019-11-15
> > > > **Thank you for your clarification.**
> > > >
> > > > I am sorry that because I edited my comments after I had submitted, there are differences between the e-mail version and the website version.
> > > >
> > > > Thank you for your quick clarification. Now I think I can deepen my understanding. I will carefully read your comments again and reconsider the problem.

---

> > > ### Author Response · Authors · 2019-11-15
> > > **One last clarification**
> > >
> > > Sorry for spamming you if this point is obvious. We would still like to make sure that there is no confusion about the following point:
> > >
> > > > "...and there are no fear of false positives..."
> > >
> > > That is only true for the naive method (that is no longer in the revised paper and that only served for the hypothetical thought experiment) which gets a lot of samples from the data-generating distribution and very cautiously only predicts those that it has seen. Of course, in reality we only have one sample from the data-generating distribution (the sample is the training KG) and we try to generalize by fitting a distribution parameterized by vector embeddings, so we have to worry about both false positives and false negatives.
> > >
> > > Thanks again for engaging in the discussion!

---

> > ### Comment · AnonReviewer3 · 2019-11-14
> > **Response to authors' response**
> >
> > - On the choice of the estimator
> >
> > > Since these methods also maximize  log-likelihood which corresponds to minimizing the discrepancy (KL-divergence) of the data-generating and the modelled distribution, we need to bound the error in terms of these (else our analysis would not apply)
> >
> > I understand the point the authors wanted to address.
> >
> >
> > - On symmetry of  $\mathcal{R}$ and $\mathcal{O}^2$.
> >
> > > On the other hand, it is not true that $\mathcal{R}$ and $\mathcal{O}^2$ are mathematically symmetric.
> >
> > I understand that these are not symmetric strictly speaking because we only need $|\mathcal{O}|$ representations for $\mathcal{O}^2$.

---

### Official Review · AnonReviewer2 · 2019-10-25
**Official Blind Review #2**

**Rating:** 3

**Review:**

This work conducts a theoretical study of the generalization bounds for the number of wrongly predicted triples of KG embedding methods. Under a "``missing completely at random'' assumption, the authors model the distributions of sampled KG and ground-truth KG, and derivate the bounds as a function of the KL-divergence by leveraging Pinsker's inequality. Preliminary discussions on how existing KG embedding methods fit into this theoretical study have been conducted.

Pros:
- This work raises a novel problem, i.e., analyzing the theoretical generalization ability of existing KG embedding methods.
- The paper is well motivated with clear writing and technically sound with rigorous formula derivation.

Cons:
1) For KG construction, people usually collect/mine in a batch/incremental fashion in which each batch focuses on certain aspects. For example, one can collect some bio facts about Persons from Wikipedia, then collect some entertainment related facts from news. So in practice, "``missing completely at random" assumption may not hold for KG.

2) The derivation is suitable for a class of KG models that maximize log-likelihood losses, while many KG embedding methods use margin-based loss functions. Although the authors mentioned "``log-loss can in principle be also used and it was observed by Trouillon & Nickel (2017) that the margin-based loss functions, used by many knowledge graph embedding methods, are more prone to overfitting compared to log-likelihood'', more rigorous theoretical analysis is suggested to verify (or refine) the applicability of the proposed analysis

3) It would be better to see more examples that applying the proposed analysis to different KG embedding methods, corresponding comparisons help to get a deeper understanding for existing KG embedding methods, and may shed further light on how to design a KG embedding method that achieves a good enough generalization with fewer model parameters.



**Experience Assessment:**

I have published in this field for several years.

**Review Assessment: Checking Correctness Of Derivations And Theory:**

I assessed the sensibility of the derivations and theory.

**Review Assessment: Checking Correctness Of Experiments:**

I assessed the sensibility of the experiments.

**Review Assessment: Thoroughness In Paper Reading:**

I made a quick assessment of this paper.

---

> ### Author Response · Authors · 2019-11-13
> **Author response**
>
> Thank you for the review!
>
> > Cons:
> > 1) For KG construction, people usually collect/mine in a batch/incremental fashion in
> > which each batch focuses on certain aspects. For example, one can collect some bio
> > facts about Persons from Wikipedia, then collect some entertainment related facts from
> > news. So in practice, "``missing completely at random" assumption may not hold for KG.
>
> First, we believe that we are transparent about this in the paper. We say that the setting corresponds to how these methods are evaluated in papers and we write “practice” (in quotes). We also mention in the conclusions that people have only recently started to look into more realistic train-test splits.
>
> Second, we also think that our analysis could help to explain the cases (if they happen) when a KGE method works well on standard train/test splits generated using a method similar to the one analyzed in our paper but fails for more realistic settings.
>
> Finally, it is impossible to obtain theoretical guarantees without any assumptions. We believe that it is vital to first analyze simpler settings such as the missing-completely-at-random assumption (which in our case corresponds to actual way people do experiments in the literature!) and only after that analyze more complex settings, such as the one suggested by the reviewer.
>
> > 2) The derivation is suitable for a class of KG models that maximize log-likelihood
> > losses, while many KG embedding methods use margin-based loss functions. Although
> > the authors mentioned "``log-loss can in principle be also used and it was observed by
> > Trouillon & Nickel (2017) that the margin-based loss functions, used by many knowledge
> > graph embedding methods, are more prone to overfitting compared to log-likelihood'',
> > more rigorous theoretical analysis is suggested to verify (or refine) the applicability of the
> > proposed analysis
>
> We agree that it would be interesting to also analyze the margin-based loss functions, but that may require different techniques and our paper has already 21 pages (including the appendix).
>
> To make sure that we do not overclaim our contribution in this direction, we added the following text to the revision:
>
> “Our results may shed further light on this latter observation, however, to verify it theoretically would require to perform a similar type of analysis as we did in this paper for the margin-based loss functions (which might require completely different techniques). Therefore we leave this comparison for future work.”
>
>
> > 3) It would be better to see more examples that applying the proposed analysis to
> > different KG embedding methods, corresponding comparisons help to get a deeper
> > understanding for existing KG embedding methods, and may shed further light on how to
> > design a KG embedding method that achieves a good enough generalization with fewer
> > model parameters.
>
> We have expanded the section on numerical examples and added SimplE in the revision.
>
> It is not difficult to apply the derived bounds on other KGE methods (as long as we can compute a bound on the Lipschitz constant which is not difficult for most methods, excluding those based on neural networks) but we are limited by the space in the main part of the document. However, if the reviewer found it crucial, we could add another to the appendix (e.g. ComplEx - but the analysis would be very similar to the one for SimplE).

---

### Decision · Program_Chairs · 2019-12-19

**Decision:**

Reject

**Comment:**

The paper provides a generalization error bound, which extends the results from PU learning, for the problem of knowledge graph completion. The authors assume a missing at random setting, and provide bounds on the triples (two nodes and an edge) that could be mistakes. Then the paper provides a maximum likelihood interpretation, as well as relations to existing knowledge graph completion methods. The problem setting is interesting, and the writing clear.

This discussion was extensive, with reviewers and authors following the spirit of ICLR and having a constructive discussion which resulted in improvements to the paper. However, there seems to be still some remaining improvements to be made in terms of clarity of presentation, as well as precision of the theoretical arguments.

Unfortunately, there are many strong submissions, and the paper as it currently stands does not satisfy the quality threshold of ICLR.